

# On the Challenges of Retrieving Phytoplankton Properties from Remote-Sensing Observations

J. Xavier Prochaska[1,2,3,4] and Robert J. Frouin[4]

[1]Affiliate of the Ocean Sciences Department, University of California, Santa Cruz
[2]Department of Astronomy & Astrophysics, University of California, Santa Cruz
[3]Kavli IPMU
[4]Scripps Institution of Oceanography, University of California, San Diego

**Correspondence:** J. Xavier Prochaska (jxp@ucsc.edu)

**Abstract.** Remote-sensing satellites provide the only means to observe the entire ocean at high-temporal resolution. Optical-sensors assess ocean color through estimates of remote-sensing reflectance ($R_{rs}(\lambda)$). We emphasize a physical degeneracy in the radiative transfer equation that relates $R_{rs}(\lambda)$ to absorption and backscattering coefficients ($a(\lambda),b_b(\lambda)$) known as inherent optical properties (IOPs). This degeneracy stems from $R_{rs}(\lambda)$ depending on the ratio $b_b(\lambda)/a(\lambda)$, preventing the independent retrieval of non-water IOPs without prior knowledge. We demonstrate that multi-spectral satellite observations lack the statistical power to recover more than three parameters describing non-water absorption and backscattering. Due to exponential-like absorption by colored dissolved organic matter and detritus at shorter wavelengths, multi-spectral $R_{rs}(\lambda)$ data cannot detect phytoplankton absorption without strict priors, leading to biased and uncertain estimates. These results challenge decades of IOP retrieval literature, including assessments of phytoplankton growth and biomass. While hyperspectral observations hold promise to recover additional parameters, significant hurdles remain in accurately quantifying IOPs and phytoplankton biomass at a global scale.

## 1 Introduction

Phytoplankton play essential roles within our ecosystem, serving as the base of the ocean food web and performing $\sim 50\%$ of all photosynthesis on Earth. Therefore, assessing phytoplankton growth and death – especially in a changing climate (Behren-feld et al., 2016; Flombaum et al., 2020) – is critical to any effort to track and predict the health of our planet. Decades of phytoplankton research have revealed significant regional variations in these process and demonstrated that phytoplankton are highly dynamic on relatively short time scales (hours to weeks, especially in coastal areas, due to tides, upwelling, pulses of freshwater inflow, and other episodic events (e.g. Cloern and Jassby, 2010). To identify any long-term trend, therefore, one must first develop a detailed picture of the variations on seasonal and shorter timescales.

Unfortunately, our ability to measure phytoplankton in-situ is greatly hampered by the vast expanse of the ocean. Measurements with high temporal frequency can only be acquired at select, fixed stations such as OceanSITES (Boss et al., 2022). Therefore, oceanographers have turned to remote-sensing satellite observations to perform high-cadence, global analyses of the ocean surface. Beginning with the Coastal Zone Color Scanner experiment (Hovis et al., 1980), multi-band observations in



optical channels have enabled the inference phytoplankton and other seawater constituent properties, such as chlorophyll-a con-

centration and absorption by colored dissolved organic matter (CDOM) and detritus (IOCCG, 2000; Siegel et al., 2002). These

properties are obtained from satellite-derived remote sensing reflectance $R_{rs}(\lambda)$, which represents the water-leaving radiance

normalized by incident solar irradiance. Their retrieval relies on recovering the absorption and backscattering coefficients $a(\lambda)$,

$b_b(\lambda)$ that govern $R_{rs}(\lambda)$ and are known as inherent optical properties (IOPs).

The underlying physics for IOP retrievals is radiative transfer: the absorption and scattering of sunlight by seawater modu-

lates and directs incident sunlight back to the satellite. While the radiative transfer physics is straightforward (but not simple;

Mobley, 2022), there are many factors that complicate the calculations. These include but are not limited to: the concentra-

tion of the constituents (typically the desired unknown), their variation with depth, the precise wavelength dependence of the

absorption and scattering coefficients of each constituent, geometric factors associated with the Sun's location relative to the

satellite. In addition, Earth's atmosphere attenuates the signal and introduces a dominant background radiation field which must

first be estimated and subtracted ("corrected") which fundamentally limits the precision of any space-based $R_{rs}(\lambda)$ estimation

(e.g. Frouin et al., 2022).

For decades, researchers have attacked this radiative transfer problem to attempt retrievals of scientifically valuable quantities

including an estimate of the phytoplankton biomass. There is a robust and well-founded literature describing (and performing)

the translation of so-called apparent optical properties (AOPs, e.g. $R_{rs}(\lambda)$) to inherent optical properties (IOPs; $a(\lambda)$, $b_b(\lambda)$)

that depend solely on the water constituents and the water itself. Ideally, one first parameterizes and then estimates ("retrieves")

the absorption and backscattering spectra of the non-water component $a_{nw}(\lambda)$, $b_{b,p}(\lambda)$ and then infers concentrations or proxies

of phytoplankton, CDOM, detritus, etc. From these, one may examine the geographic distribution and temporal evolution of

fundamental biological processes across the global ocean (e.g. Behrenfeld et al., 2005; Siegel et al., 2014; Fox et al., 2022).

During the development of a diverse set of IOP retrieval algorithms for this purpose (see Werdell et al., 2018, for a review),

the ocean optics community has acknowledged key challenges to the problem largely independent of those associated with

radiative transfer. These include uncertainties related to the atmospheric corrections, non-uniqueness between common con-

stituents (e.g. CDOM and detritus), and retrieving multiple unknowns from limited datasets (e.g. multi-spectral observations).

A few, sparsely-cited works have also highlighted a more fundamental obstacle to the process: a physical "ambiguity" in the

inversion of the radiative transfer equation (Sydor et al., 2004; Defoin-Platel and Chami, 2007). Unfortunately, this problem

has often been confused or conflated with the statistical limitations of an insufficient number of bands measuring $R_{rs}(\lambda)$

(Werdell et al., 2018; Cetinić et al., 2024). As such, while the community has acknowledged challenges to IOP retrievals from

remote-sensing observations, rigorous assessment of the algorithms themselves has been limited and usually only performed

in the context of comparisons to sparse, in-situ observations (e.g. Lee, 2006; Seegers et al., 2018).

Another fundamental aspect of the problem is that we do not know the optimal basis functions that describe $a(\lambda)$ and $b_b(\lambda)$

nor even the complete set (e.g. Garver et al., 1994). Indeed, it is an aspiration within the ocean color field to recover (or even

discover) the composition of phytoplankton communities (e.g. Mouw et al., 2017). The ocean color research community has

hoped that the main limitation is the sparsity of existing multi-spectral bands provided by current satellites and that hyperspec-

tral observations will lead to a major breakthrough. Indeed, Cael et al. (2023) has demonstrated from a data-driven analysis of




$R_{rs}(\lambda)$ spectra its limited information content, i.e. only $\sim 2$ degrees-of-freedom in multi-spectral, satellite observations. But
they also concluded that in-situ hyperspectral datasets provide only one or two addition degrees of freedom to describe the
seawater composition. In this manuscript, we examine this question from a new angle – with the standard approach of IOP
retrievals – and reach similar conclusions.

Here, we introduce the Bayesian INferences with Gordon coefficients (BING) package for ocean retrievals in a Bayesian
context (see Erickson et al., 2020, 2023, for a complementary Basyesian approach). Our approach follows many of the standard
assumptions of widely adopted algorithms in the literature, e.g. the generalized IOP (GIOP) model (Werdell et al., 2013),
the Garver-Siegel-Maritorena (GSM) algorithm (Maritorena et al., 2002). In addition, we emphasize and expand upon the
"ambiguity" problem – a physical degeneracy in the radiative transfer equation that couples reflectances to IOPs – which
fundamentally limits IOP retrievals. In turn, we demonstrate that IOP retrievals from multi-spectral datasets constrain at most
three parameters describing $a_{\mathrm{nw}}(\lambda)$ and $b_{b,p}(\lambda)$. Consequently, if the spectral shape of CDOM absorption is allowed to vary
and remains unconstrained, it becomes challenging, even impossible, to independently retrieve phytoplankton absorption with
high confidence. This limitation applies to previous satellite missions equipped with multi-band sensors, emphasizing the need
for additional constraints or improved observational capabilities for more accurate phytoplankton absorption retrievals. We
then examine the prospects for IOP retrievals with hyperspectral observations and discuss additional opportunities to address
the deep degeneracies that lurk within.

Therefore, if one allows for the fact that the spectral shape of CDOM absorption varies and is unknown, then we show
one cannot retrieve a measurement of phytoplankton absorption. This includes all previous missions with satellites carrying
multi-bands sensors. We then examine the prospects for IOP retrievals with hyperspectral observations and discuss additional
opportunities to address the deep degeneracies that lurk within.

## 2 Methods

### 2.1 Bayesian Formalism

At the heart of our analysis is an open-source Bayesian inference algorithm developed for the retrieval of IOPs from remote
sensing reflectances, the BING package. The primary motivations for introducing a Bayesian framework are twofold: (i) it
forces one to explicitly describe all of the priors that influence the result; (ii) it leverages well developed techniques to assess
error and correlations in the results without requiring Gaussianity, i.e. the assumption that errors, uncertainties, or distributions
of retrieved parameters follow a Gaussian (normal) distribution; and (iii) it permits well-established approaches for performing
model selection, i.e., estimating the maximum number of free parameters one can use to describe the data. BING is conceptually
similar to the algorithm presented in Erickson et al. (2020) to analyze diatoms and *Noctiluca scintillans*, although BING adopts
a Monte Carlo Markov Chain sampler. In addition, the code base incorporates several previous inference algorithms (e.g. GSM,
GIOP) and is extensible to include any new, user-driven parameterization of the IOPs. The BING package is purely Python and
is available on GitHub (Prochaska, 2024).



Provided a forward model (described in the following section) and a parameterization of $a(\lambda)$ and $b_b(\lambda)$, the Bayesian inference is straightforward and a wealth of well-trodden approaches and software packages are available. For BING, we adopt the Monte Carlo Markov Chain (MCMC) formalism which empirically derives the posterior probabilities for the $a(\lambda)$, $b_b(\lambda)$ parameterization $P(X|Y)$ including full uncertainties and all of the cross-correlation terms. This requires the definition of a

likelihood function $P(Y|X)$, which will have the form:

$$P(Y|X) \propto \exp\left\{-\frac{1}{2}[Y - H(X)]^{\mathrm{T}}C[Y - H(X)]\right\} \tag{1}$$

where $Y$ represents the measured $R_{rs}(\lambda)$ values, $C$ is the full covariance matrix of $R_{rs}(\lambda)$ including correlations, and $H(X)$ is the forward model of $R_{rs}(\lambda)$ at the locations of $Y$.

It can be shown that an MCMC analysis converges to the exact solution if run for an infinitely long time; in practice, the

calculations tend to converge after $\approx 10,000$ iterations. For the analysis here, we generally run for 75,000 trials with at least 2 walkers per parameter (and at least 16 walkers) and only analyze the last 7,000 iterations of each. This release of BING uses the EMCEE sampler (Foreman-Mackey et al., 2013) which was developed for astrophysical applications and has seen wide-spread adoption in the field (over 8,000 citations).

We have also implemented standard $\chi^2$ minimization (Levenberg-Marquardt; L-M) as a fitting option to speed-up model

development and portions of the analysis. This also enables one to implement standard inference models in the literature (e.g. GSM and GIOP) which generally use L-M optimization.

Note that the Bayesian approach in BING involves using Bayes' theorem to update the probability of a hypothesis based on prior knowledge and new data. When retrieving IOPs from $R_{rs}(\lambda)$, this approach explicitly incorporates all available prior information about the IOPs and their uncertainties into the model. By doing so, it allows for a more transparent and rigorous

estimation process. The Bayesian framework considers the likelihood of the observed $R_{rs}(\lambda)$ given the IOPs and combines it with the prior probability distributions of the IOPs to obtain a posterior distribution. This posterior distribution provides a probabilistic solution to the inverse problem, highlighting the most likely values of the IOPs while quantifying the uncertainties, leading to more reliable and informed retrievals.

In reference to the detection of phytoplankton, we consider a series of models without and with $a_{\mathrm{ph}}(\lambda)$ absorption. We then

examine the evidence for models with $a_{\mathrm{ph}}(\lambda)$ in the Bayesian context by applying standard approaches for model selection, i.e., assessing the balance between model fit and complexity. Specifically, we have evaluated the Aikake and Bayesian information criteria (AIC,BIC) which are akin to $\chi^2$-difference tests (Bentler and Bonett, 1980):

$$\mathrm{AIC} = 2k - 2\ln\mathcal{L} \;, \tag{2}$$

and

$$\mathrm{BIC} = k\ln n - 2\ln\mathcal{L} \;, \tag{3}$$





with $n$ the number of $R_{rs}(\lambda)$ measurements and $\mathcal{L}$ the likelihood function calculated assuming Gaussian statistics for uncertainties $\sigma(R_{rs}(\lambda))$. The likelihood function $\mathcal{L}$ quantifies the probability of observing the data given the specific forward model and its parameters. Since it is calculated under the assumption of Gaussian (normal) statistics for the $R_{rs}(\lambda)$ uncertainties, this means that the observed data, i.e., the $R_{rs}(\lambda)$ measurements, are assumed to be normally distributed around the model predictions with the standard deviation. For our hyperspectral analysis where $n \gg 10$, the BIC offers a more stringent constraint but the results with AIC are qualitatively similar. A high AIC or BIC value implies that the model is less likely to be the best model given the data, considering both the fit and complexity. Quantitatively, we assess model selection by evaluating the difference in BIC for any two models

$$\Delta\mathrm{BIC_{i,j}} = \mathrm{BIC_i} - \mathrm{BIC_j} \ , \tag{4}$$

where $\Delta\mathrm{BIC_{i,j}} < 0$ indicates that model $i$ is preferred and vice versa.

## 2.2 Radiative transfer with a physical degeneracy

To construct any such algorithm, one must have a well-defined forward model to predict the observables, here remote-sensing reflectances $R_{rs}(\lambda)$. For IOP inversion, this means a radiative transfer model – or its approximation – which estimates $R_{rs}(\lambda)$ from $a(\lambda)$ and the backscattering coefficients $b_b(\lambda)$. The majority of IOP retrieval algorithms developed by the community have used the quasi single-scattering approximation (QSSA) originally introduced by Hansen (1971) and translated to ocean color by Gordon (1973) (see also Zege et al., 1991). This approach was refined further by Gordon (1986) who approximated the sub-surface remote reflectances $r_{rs}(\lambda)$ with a Taylor series expansion:

$$r_{rs}(\lambda) = \sum_{i=1}^{N} G_i\, u(\lambda)^i \ , \tag{5}$$

with

$$u(\lambda) \equiv \frac{b_b(\lambda)}{a(\lambda) + b_b(\lambda)} \ . \tag{6}$$

Most IOP retrieval algorithms have taken $N = 2$ and set the coefficients as constants $G_1 = 0.0949$ and $G_2 = 0.0794$, i.e. values independent of wavelength. In this manuscript and for the default mode of BING, we adopt the same prescription and coefficients, and scrutinize the accuracy of this assumption in Supp A. For the results in the main text, we assume a perfect forward model, i.e. we use Equation 5 to generate the target $r_{rs}(\lambda)$ and perform the fits on these values. In practice, we work with remote-sensing reflectances $R_{rs}(\lambda)$ following a standard conversion from $r_{rs}(\lambda)$ (Lee et al., 2002):

$$r_{rs}(\lambda) = \frac{R_{rs}(\lambda)}{0.52 + 1.17 R_{rs}(\lambda)} \tag{7}$$





Despite the approximation of Equation 5, it does capture a salient aspect of the physics: the functional dependence of $R_{rs}(\lambda)$ on $u(\lambda)$ and thereby the IOPs $a(\lambda)$ and $b_b(\lambda)$. However, this dependence reveals an especially challenging aspect of IOP retrievals: because $u(\lambda)$ is a function of the ratio of $b_b(\lambda)/a(\lambda)$,

150
$$r_{rs}(\lambda) = Func\left(\frac{b_b}{a}\right) \ , \tag{8}$$

the radiative transfer solutions are *physically degenerate* in $b_b/a$. Put succinctly, any IOP solution that recovers a set of $R_{rs}(\lambda)$ observations can be replaced by an infinite set that preserves the $b_b/a$ ratio. Therefore, the retrieval is only tractable if one implements strong constraints (known as priors in Bayesian analysis) on the functional forms of $a(\lambda)$ and $b_b(\lambda)$. In section 3.1, we examine the consequences of this physical degeneracy on IOP retrievals.

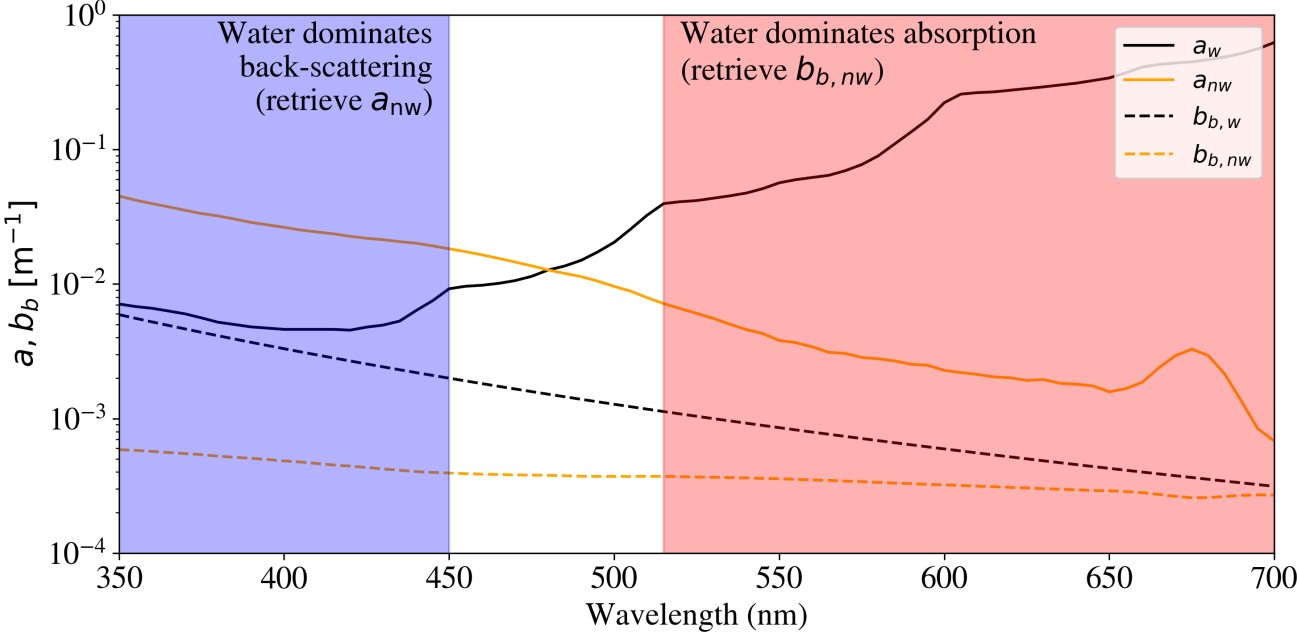

**Figure 1.** Comparison of the IOP spectra for water ($a_{\mathrm{w}}$, $b_{b,\mathrm{w}}$; black solid and dashed curves) against one example set of non-water spectra ($a_{\mathrm{nw}}$, $b_{b,p}$; orange solid and dashed curves) from L23 (their index=170, an example representative of the open ocean with low $Chla$ concentration). The red region indicates where absorption by water dominates ($a_{\mathrm{w}} > 5a_{\mathrm{nw}}$). In this region, reflectance measurements constrain the non-water component of backscattering. Similarly, the blue region is where water dominates backscattering and retrievals may constrain $a_{\mathrm{nw}}$. In turn, for observations with noise, the inversion problem has very limited constraints on the non-water components.



## 2.3 A Hyperspectral IOP Dataset

For the development and testing of BING, we have leveraged a hyperspectral dataset of $a(\lambda)$,$b_b(\lambda)$ spectra made public by Loisel et al. (2023) (hereafter L23). The spectra sample waters of both Case I and Case II properties with chlorophyll-a concentrations varying from $Chla \approx 0.01 - 10\,\mathrm{mg\,m^{-3}}$. We use their $X = 4, Y = 0$ model which includes inelastic scattering (not relevant here given our strict adoption of Equation 5 throughout) and the Sun at the zenith. The IOP spectra were generated from their database of in-situ measurements of phytoplankton $a_{\mathrm{ph}}(\lambda)$ and models of several additional constituents: CDOM $a_{\mathrm{g}}(\lambda)$, pure seawater $a_{\mathrm{w}}(\lambda)$, and detritus $a_{\mathrm{d}}(\lambda)$. L23 then generated estimates of the backscattering coefficients $b_{b,p}(\lambda)$ following standard assumptions based on in-situ and laboratory work (see Loisel et al., 2023, for additional details). These 3,320 $a(\lambda)$ and $b_b(\lambda)$ spectra define our dataset, and while they range from 350-750 nm we restrict analysis to $\lambda = 400 - 700$ nm.

Although IOPs retrievals are greatly challenged by the physical degeneracy in the radiative transfer described in the previous section, a positive aspect of the problem is the presence of water which introduces an ever-present and precisely known constraint on the problem (except in the ultraviolet, $\lambda < 400$ nm, Mason et al. (2016)). The absorption $a_{\mathrm{w}}(\lambda)$ and backscattering $b_{b,w}(\lambda)$ spectra of pure seawater impose priors on the model that serve to partially alleviate the physical degeneracy described in the previous section. First, $a_{\mathrm{w}}(\lambda)$ and $b_{b,w}(\lambda)$ span the entire spectrum and therefore couple the otherwise independent $R_{rs}(\lambda)$ values. Second, to the extent that the shapes of $a_{\mathrm{w}}$ and $b_{b,w}$ are unique relative to other constituents this helps one avoid the $b_b/a$ degeneracy. Third, the strong absorption of water at $\lambda > 500$ nm and the relatively high magnitude of $b_{b,w}(\lambda)$ at $\lambda < 450$ nm define regions where one may retrieve information on the non-water components (Sydor et al., 2004).

On the last point, Figure 1 compares the absorption and backscattering coefficients of water against one example of non-water spectra $a_{\mathrm{nw}}(\lambda)$, $b_{b,p}(\lambda)$ from the L23 dataset. As emphasized in the Figure, at red wavelengths $a \approx a_{\mathrm{w}}(\lambda)$ and $b_{b,w} \approx b_{b,p}$ such that the observations may constrain $b_{b,p}$. Similarly at $\lambda < 450$ nm, $b_b \approx b_{b,w}$ and $a_{\mathrm{nw}}(\lambda) > a_{\mathrm{w}}(\lambda)$ such that the observations constrain $a_{\mathrm{nw}}$. These inferences from Figure 1, however, rely on the strong (but frequently satisfied) prior that $a_{\mathrm{w}}(\lambda) > a_{\mathrm{nw}}(\lambda)$ at $\lambda > 500$ nm and $b_{b,w}(\lambda) > b_{b,p}(\lambda)$ at $\lambda < 450$ nm. If this is relaxed, e.g. if $a_{\mathrm{nw}}(\lambda)$ and $b_{b,p}(\lambda)$ may take on *any* values then the $b_b/a$ degeneracy forces an infinite set of solutions (i.e. no unique retrieval is possible, see Section 3.1).

## 2.4 Simulating Satellite Observations

For many of the analyses presented here, we have generated simulated $R_{rs}(\lambda)$ spectra for several multi-spectral and hyperspectral missions. For the fits presented in this manuscript, we ignore the $R_{rs}(\lambda)$ spectra provided by L23 (generated with Hydrolight) and instead use Equations 5-7 to calculate $R_{rs}(\lambda)$ from $a(\lambda)$ and $b_b(\lambda)$. We then resample these spectra to the bands/channels of several satellite missions:

MODerate resolution Imaging Spectroradiometer (MODIS)/Aqua: We adopt 8 multi-spectral bands as listed in Table 1 corresponding to MODIS/Aqua and we evaluate $R_{rs}(\lambda)$ at the center of each. For uncertainties, we have estimated the RMS difference between satellite and in-situ $R_{rs}(\lambda)$ "match-up" measurements collated on the SeaBASS database (Werdell and Bailey, 2002) after iteratively clipping any $4\sigma$ outliers. Figure 2 shows an example of the data and clipping for one band. We





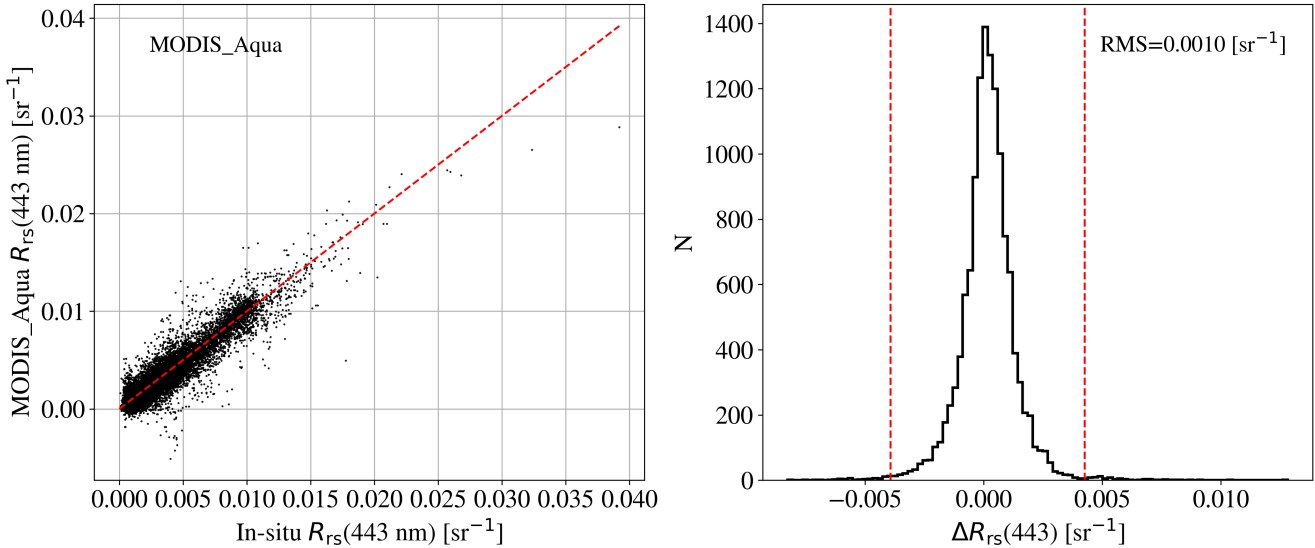

**Figure 2.** (left) Comparison of the MODIS measurements of $R_{rs}(\lambda)$ at $\lambda = 443$ nm against in-situ observations at the same wavelength. These were taken from the SeaBASS site (Werdell and Bailey, 2002) dedicated to MODIS matchups (NASA Goddard Space Flight Center, 2024). The points follow the over-plotted one-to-one line relatively well, albeit with significant scatter which we assess as the RMS noise in the MODIS observations. (right) Distribution of the difference between in-situ and satellite $\Delta R_{rs}(\lambda) = R_{rs}(\lambda)^{\mathrm{in-situ}} - R_{rs}(\lambda)^{\mathrm{MODIS}}$. The red dashed-lines show the $4\sigma$ interval beyond which we clipped the data when calculating the noise estimate (RMS).

**Table 1.** MODIS Data

| Band | $\sigma(R_{rs}(\lambda))$ |
|---|---|
| (nm) | (sr$^{-1}$) |
| 412 | 0.0012 |
| 443 | 0.0009 |
| 488 | 0.0008 |
| 531 | 0.0007 |
| 547 | 0.0007 |
| 555 | 0.0007 |
| 667 | 0.0002 |
| 678 | 0.0001 |

Notes: The error has assumed that $1/2$ of the variance is due to the in the in-situ measurements.

further assumed that one half of the variance is due to the in-situ observations themselves. These RMS values are also provided in Table 1, and we find they are in good agreement with other estimations (Zhang et al., 2022; Kudela et al., 2019).

Sea-viewing Wide Field-of-view Sensor (SeaWiFS)/SeaStar: We followed a similar procedure for SeaWiFS using 6 bands and

190    the uncertainties provided in Table 2.




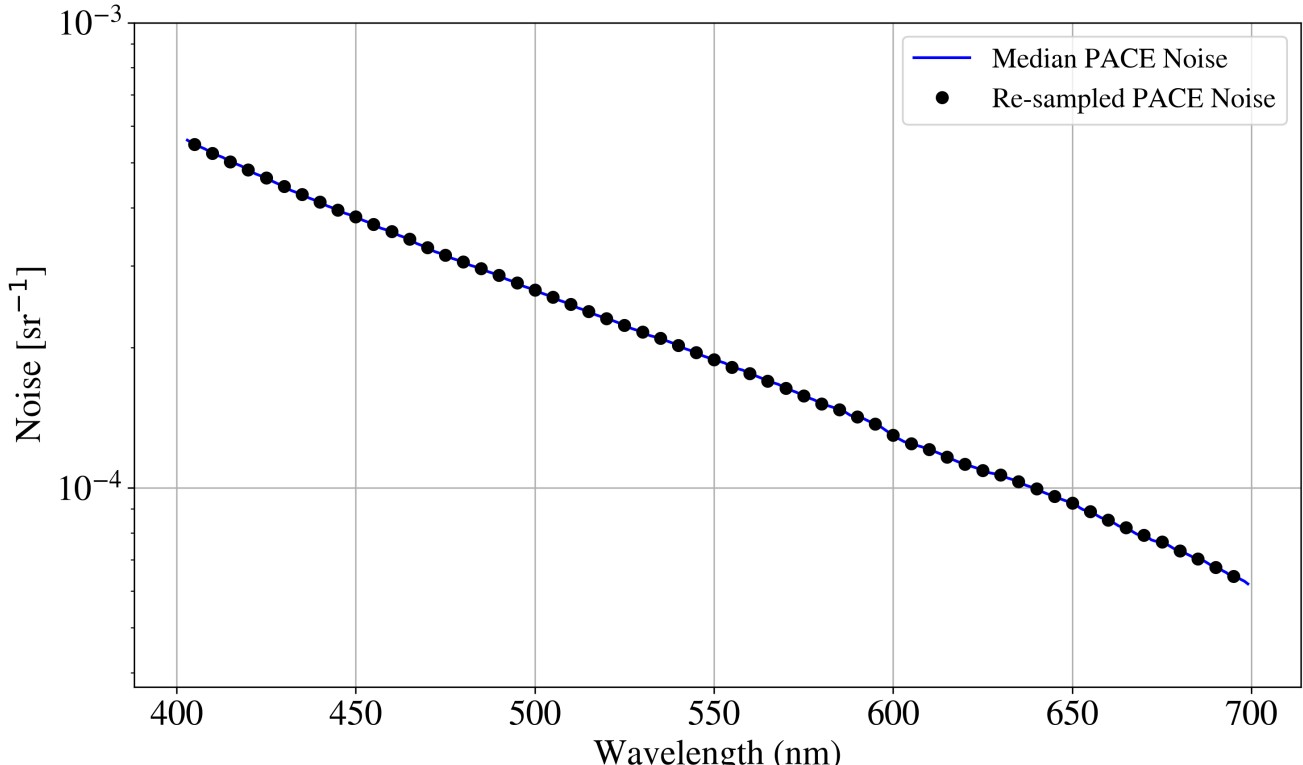

**Figure 3.** The blue curve is the median PACE uncertainty in $R_{rs}(\lambda)$ estimated in the Level 2 product, v2.0 for one granule ( PACE_OCI.20240413T175656.L2.OC_AOP.V2_0.NRT.nc). The black dots are the values of $\sigma(R_{rs}(\lambda))$ adopted in this manuscript when simulating PACE spectra with noise.

**Table 2.** SeaWiFS Data

| Band (nm) | $\sigma(R_{rs}(\lambda))$ (sr$^{-1}$) |
|---|---|
| 412 | 0.0014 |
| 443 | 0.0011 |
| 490 | 0.0009 |
| 510 | 0.0006 |
| 555 | 0.0007 |
| 670 | 0.0003 |

Notes: The error has assumed that 1/2 of the variance is due to the in the in-situ measurements.

Ocean Color Instrument (OCI)/Plankton, Aerosol, Cloud, ocean Ecosystem (PACE) Satellite: For simulated OCI spectra, we assumed $\delta = 5$ nm sampling and limited the wavelength range from $400 - 700$ nm. The lower bound is due to (i) greater



uncertainties in the atmospheric corrections, (ii) greater uncertainty in water absorption and scattering, (iii) greater uncertainty in how best to parameterize the non-water components in the UV. The upper wavelength bound is to avoid systematics that likely dominate the uncertainty at the lowest $R_{rs}(\lambda)$ signals. Furthermore, we have limited measurements on the absorption of standard seawater constituents (e.g. phytoplankton) at these longer wavelengths.

For the PACE noise model, we downloaded a single granule (2,175,120 pixels) of Level 2 data, v2.0: PACE_OCI.20240413T175656.L2.OC_AOP.V2_0.NRT.nc. We then took the median uncertainty spectrum (Rrs_unc, Zhang et al., 2022) for all non-flagged data between 33-40N and 73-78W. This median uncertainty spectrum is plotted in Figure 3 at the Level 2 wavelength sampling ($\approx 2.5\,\mathrm{nm}$). We also show the values adopted at our $\delta = 5\,\mathrm{nm}$ sampling, and one notes that we did not adjust $\sigma(R_{rs}(\lambda))$ despite the larger sampling size. This is because OCI is over-sampled at $\delta = 2.5\,\mathrm{nm}$, i.e. neighboring data points are highly correlated. Further work should assess the degree of this correlation to obtain a better noise estimate.

Note that the noise is independent across spectral bands, meaning no spectral correlation is assumed. However, in standard atmospheric correction procedures, noise is expected to be spectrally correlated due to systematic uncertainties in aerosol and Rayleigh scattering treatments, as well as instrumental effects. Ignoring these correlations could introduce additional uncertainties in the retrievals by misrepresenting the spectral structure of the observed signals. Whether treating noise as uncorrelated maximizes or underestimates the effective retrieval uncertainty depends on the interplay between error propagation and the constraints imposed by the retrieval algorithm. A more rigorous treatment should incorporate spectral noise correlations to provide a more accurate error characterization.

## 2.5 IOP Models

For the principle analysis of this manuscript, we will consider a series of increasingly complex models for the IOPs $a_{\mathrm{nw}}(\lambda)$ and $b_{b,p}(\lambda)$. We generally follow common practice for models of $a_{\mathrm{nw}}(\lambda)$ and $b_{b,p}(\lambda)$ which have been informed by in-situ and laboratory measurements of ocean constituents. In turn, we will examine the maximum complexity that can be statistically constrained by observations designed to mimic satellite retrievals, e.g. data from multi-band and hyperspectral observations.

Consider first the simplest scenario we may conceive: a two-parameter $[k = 2]$ model with both $a_{\mathrm{nw}}(\lambda)$ and $b_{b,p}(\lambda)$ taken as constant at all wavelengths:

$$a_{\mathrm{nw}}(\lambda) \quad = A_{cst} \tag{9}$$

$$b_{b,p}(\lambda) \quad = B_{nw} \tag{10}$$

This model may not have physical merit, but it serves as a baseline for comparison with other, physically motivated IOP scenarios.

Now consider three additional models of increasing complexity. The $[k = 3]$ model,





$$a_{\mathrm{nw}}(\lambda) = A_{dg} \exp[-S_{dg}(\lambda - 400)] \tag{11}$$

$$b_{b,p}(\lambda) = B_{nw} \tag{12}$$

where $\lambda$ is expressed in nm and where one assumes the non-water absorption is strictly an exponential function. This spectral shape is commonly used to describe the absorption by CDOM and/or detritus. In-situ absorption measurements show typical values of $S_{dg} \approx 0.015$ for CDOM (Roesler et al., 1989) and $S_{dg} \approx 0.010$ for detritus (Stramski et al., 2001). Our fiducial models only require $S_{dg} > 0$ but we also consider stricter priors on this parameter.

The $[k = 4]$ model

$$a_{\mathrm{nw}}(\lambda) = A_{dg} \exp[-S_{dg}(\lambda - 400)] \tag{13}$$

$$b_{b,p}(\lambda) = B_{nw} (\lambda/600)^{\beta_{nw}} \tag{14}$$

which adds the commonly adopted power-law for backscattering by particulate matter (Gordon and Morel, 1983). For the $[k = 4]$ and $[k = 5]$ models we allow $\beta_{nw}$ to vary but consider fixed exponents for other models introduced below (and, strictly speaking, $\beta_{nw}=0$ for models $[k = 2]$ and $[k = 3]$).

Last in this sequence, the $[k = 5]$ model includes a phytoplankton component

$$a_{\mathrm{nw}}(\lambda) = A_{dg} \exp[-S_{dg}(\lambda - 400)] + A_{ph} a_{\mathrm{ph}}(\lambda) \tag{15}$$

$$b_{b,p}(\lambda) = B_{nw} (\lambda/600)^{\beta_{nw}} \tag{16}$$

with $a_{\mathrm{ph}}(\lambda)$ introduced to capture "typical" absorption by phytoplankton. It is expected and observed that this component may exhibit the greatest complexity. Indeed, scientifically the community aims to distinguish the potentially large variations in phytoplankton families throughout the ocean and inland waters. For this $[k = 5]$ model, we adopt the parameterization of Bricaud et al. (1995):

$$a_{\mathrm{ph}}(\lambda) = A_{\mathrm{ph}}(\lambda) [Chla]^{E_{\mathrm{ph}}(\lambda)} \tag{17}$$

with $Chla$ the Chlorophyll-a concentration in $\mathrm{mg\,m^{-3}}$ and the tabulation of $A_{\mathrm{ph}}(\lambda)$ and $E_{\mathrm{ph}}(\lambda)$ are provided by Bricaud et al. (1998). Furthermore, we follow L23 (and previous literature) and assume

$$Chla = a_{\mathrm{ph}}(440\,\mathrm{nm})/0.05582\,\mathrm{m^{-1}} \tag{18}$$

so that phytoplankton absorption is described by one free parameter: $a_{\mathrm{ph}}(440\,\mathrm{nm})$ (aka $A_{ph}$).



For these models, we impose the following priors (constraints) on the 5 parameters. For each amplitude ($A_{dg}$, $A_{ph}$, $B_{nw}$), we assume a uniform log prior from $10^{-6}$ to $10^5\,\mathrm{m}^{-1}$ in magnitude. For the shape parameters, we assume a uniform prior for $S_{dg}$ in the interval $\mathcal{U} = [0.1, 0.2]$ and that $\beta_{nw}$ has a uniform prior with values $\mathcal{U} = [0, 2]$. These priors for the shape parameters are motivated by the range of measured in-situ values for each (Roesler et al., 1989; Lee et al., 2002).

For a portion of the analysis, we consider the GIOP and GSM models which have been widely adopted within the community, including operational implementations by NASA. These two models are effectively constrained versions of the $[k = 5]$ model with different priors on the parameters. In particular, both GSM and GIOP adopt a fixed $S_{dg} - 0.018\mathrm{nm}^{-1}$ for GIOP and $0.0206\mathrm{nm}^{-1}$ for GSM. The models also either adopt a fixed value for $\beta_{nw}$ (1.0337 for GSM) or estimate it from the $R_{rs}(\lambda)$ measurements (GIOP) with a separate prescription (Lee et al., 2002).

Each model also has a different approach for setting the shape of $a_{\mathrm{ph}}(\lambda)$ than our $[k = 5]$ model. The standard GIOP model estimates $Chla$ from a separate prescription (typically the OC4 algorithm from (O'Reilly et al., 1998) and then adopts Equation 15 for the shape of $a_{\mathrm{ph}}(\lambda)$. For GSM, we adopt their multi-spectral description of $a_{\mathrm{ph}}(\lambda)$ and interpolate to hyperspectral resolution as needed.

We consider one final scenario, a many-parameter model ($[k = \mathrm{free}]$) with one free parameter per wavelength channel for each of $a_{\mathrm{nw}}(\lambda)$ and $b_{b,p}(\lambda)$. This model is used to attempt retrievals with any arbitrary shape for the IOPs.

## 3 Results

### 3.1 Failed Attempts at Arbitrary IOP Retrievals

The physical degeneracy in the radiative transfer relating $R_{rs}(\lambda)$ to IOPs (Section 2.2) implies that the greater the freedom that one allows for $a_{\mathrm{nw}}(\lambda)$ or $b_{b,p}(\lambda)$, the more degenerate the solutions. This fundamentally limits our ability to retrieve arbitrary $a(\lambda)$ or $b_b(\lambda)$ even in the presence of *perfect* data (infinite number of channels and no uncertainty). Therefore, no algorithm can retrieve arbitrary or even highly complex $a(\lambda)$ and $b_b(\lambda)$. To make progress, one most also impose strong constraints on $a_{\mathrm{nw}}(\lambda)$ and $b_{b,p}(\lambda)$ to recover unique or most probable solutions. These priors, however, must ensure that the values of $a$ and $b_b$ cannot vary freely at any individual wavelength where one seeks a retrieval in a way which holds their ratio constant.

To demonstrate this with an example, we performed a series of IOP retrievals of $b_{b,p}(\lambda)$ assuming a perfect forward model (Equation 5), perfect knowledge of the uncertainties, and perfect knowledge of water ($a_{\mathrm{w}}$, $b_{b,w}$). In this case, we solve for "arbitrary" $a_{\mathrm{nw}}(\lambda)$ and $b_{b,p}(\lambda)$ by parameterizing each with 61 free parameters, one per wavelength channel of the hyperspectral data. We show the fits to $R_{rs}(\lambda)$ for the index=170 spectra of L23 in Figure 4 assuming the exact answer and then $a_{\mathrm{nw}}(\lambda)$ with a series of assumed scale factors. We then solved for the corresponding $b_{b,p}(\lambda)$ spectra that provide the same best-fit to $R_{rs}(\lambda)$. Indeed, there are an infinite number of $a_{\mathrm{nw}}(\lambda)$, $b_{b,p}(\lambda)$ solutions; one cannot retrieve arbitrary IOPs from $R_{rs}(\lambda)$ spectra.

This physical degeneracy limits the information content of retrievals and precludes arbitrary functional forms for $a_{\mathrm{nw}}(\lambda)$ and $b_{b,p}(\lambda)$, e.g. models which strive to retrieve arbitrary $a_{\mathrm{nw}}(\lambda)$ are ruled out. Instead, one must impose constraints (priors) on the functional forms of $a_{\mathrm{nw}}(\lambda)$ and $b_{b,p}(\lambda)$ (i.e. parameterize them) and, ideally, priors on the parameters themselves.





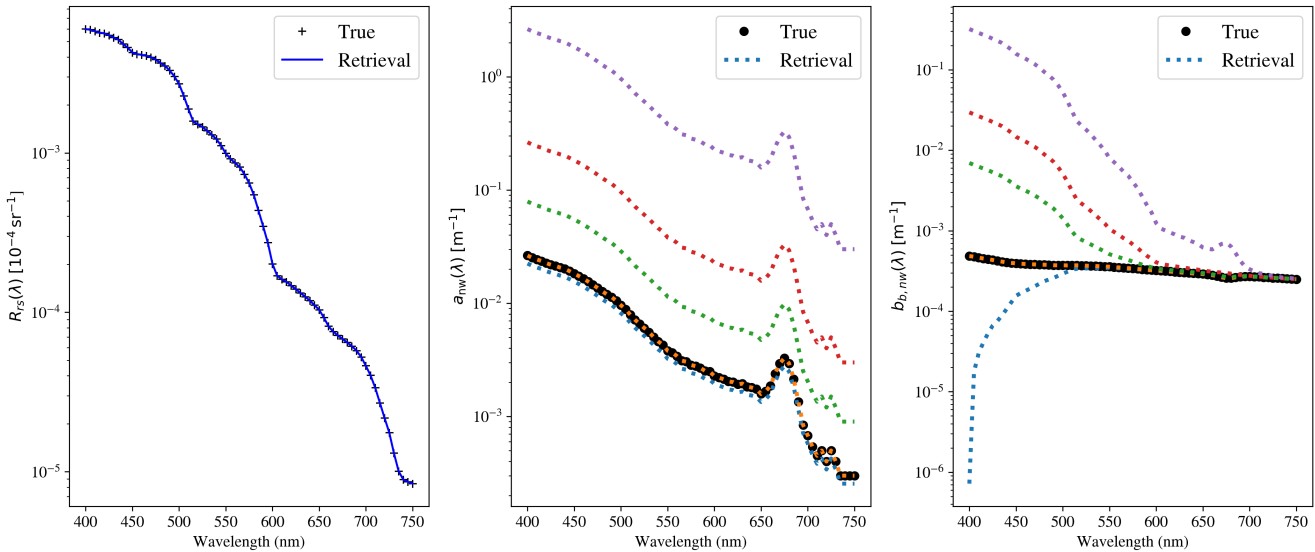

**Figure 4.** Example of a series of $R_{rs}(\lambda)$ "fits" to a $R_{rs}(\lambda)$ spectrum (left) for varying $a_{\mathrm{nw}}(\lambda)$ (center) and $b_{b,p}(\lambda)$ (right). In this example, we parameterized $a_{\mathrm{nw}}(\lambda)$ and $b_{b,p}(\lambda)$ each with 61 free parameters, one for every wavelength channel. We then altered the non-water absorption spectrum by a range of scaling factors (0.9, 3., 10., 100.) and derived the $b_{b,p}(\lambda)$ spectrum that maintaine a perfect solution for $R_{rs}(\lambda)$. Owing to the physical degeneracy in the radiative transfer equation, $R_{rs}(\lambda) = F(a/b_b)$, there are an infinite number of such solutions yielding an infinite uncertainty in $a_{\mathrm{nw}}(\lambda)$ and $b_{b,p}(\lambda)$.

## 3.2 Parameterized IOP Retrievals with BING

We now perform attempted retrievals of the IOPs $a_{\mathrm{nw}}(\lambda)$ and $b_{b,p}(\lambda)$ using assumed spectral shapes with a set of increasingly

complex prescriptions. We begin by fitting the $[k=2]$ model to two examples from the L23 dataset (Figure 5): one chosen to be representative of their full dataset and the other chosen to have a higher than typical phytoplankton absorption $a_{\mathrm{ph}}(\lambda)$ relative to the combined CDOM and detritus components $a_{\mathrm{dg}}(\lambda)$ (i.e. $a_{\mathrm{ph}}(440\,\mathrm{nm}) > a_{\mathrm{dg}}(440\,\mathrm{nm})$). Both have relatively low $Chla$ concentrations ($\approx 0.1\,\mathrm{mg\,m^{-3}}$). For these, we fit to $R_{rs}(\lambda)$ values calculated directly from Equation 5 and assume a constant S/N=50 for the $R_{rs}(\lambda)$ values for the likelihood calculation. Despite the extreme simplicity of this $[k=2]$ model,

the $R_{rs}(\lambda)$ fits are not too dissimilar from the true values, especially for the high $a_{\mathrm{ph}}(\lambda)/a_{\mathrm{dg}}(\lambda)$ example, and we note that this $a_{\mathrm{ph}}$-dominated spectrum has a low total non-water absorption, i.e. weak $a_{\mathrm{dg}}$ absorption. This follows from our discussion of Figure 1: water backscattering and absorption dominates the solution at short and long wavelengths respectively and the non-water components have limited impact on the $R_{rs}(\lambda)$, i.e. we primarily measure seawater from IOP retrievals especially for ocean waters with low chlorophyll concentrations.

Figure 5 shows the best solutions derived with BING for the $[k=2-5]$ models. While the log-scaling of the $R_{rs}(\lambda)$ panels hides differences at the few percent level, it emphasizes that distinguishing between models requires nearly perfect observations. Notably, when $k=2$, the model does not fit the observed $a_{\mathrm{nw}}(\lambda)$ and $b_{b,p}(\lambda)$ well, suggesting that this level



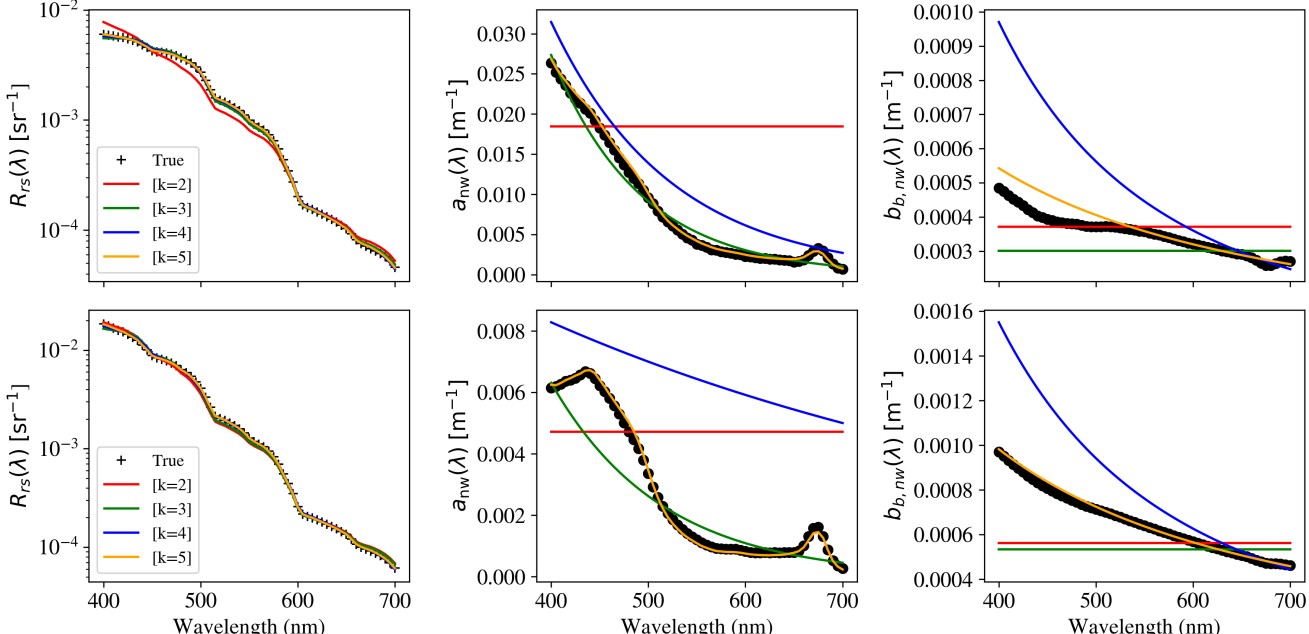

**Figure 5.** Retrievals of IOPs – (middle panels) $a_{\mathrm{nw}}(\lambda)$ and (right panels) $b_{b,p}(\lambda)$ from fits to (left panels) remote-sensing reflectances $R_{rs}(\lambda)$ assuming perfect radiative transfer and without adding noise. The black points are the true values of $R_{rs}(\lambda)$, $a_{\mathrm{nw}}(\lambda)$, and $b_{b,p}(\lambda)$ for two examples from the L23 dataset: (index=170) which we chose as representative of the full dataset and (index=1032) which has $a_{\mathrm{ph}} > a_{\mathrm{dg}}$ at 440 nm. The figure shows solutions for a series of models with increasing complexity and number of free parameters $k$. These models are ([$k = 2$]; red) constant $a_{\mathrm{nw}}(\lambda)$ and $b_{b,p}(\lambda)$; ([$k = 3$]; green) constant $b_{b,p}(\lambda)$ and a two-parameter exponential for $a_{\mathrm{nw}}(\lambda)$; ([$k = 4$]; blue) exponential $a_{\mathrm{nw}}(\lambda)$ and a power-law for $b_{b,p}(\lambda)$; and ([$k = 5$]; orange) power-law $b_{b,p}(\lambda)$ and $a_{\mathrm{nw}}(\lambda)$ modeled by the exponential and a phytoplankton function (see text for further details). It is evident that all of the $k \geq 3$ models produce excellent fits to the $R_{rs}(\lambda)$ data at the few percent level.

of complexity is insufficient to fully capture the IOP variability. However, despite these discrepancies in IOPs, the $R_{rs}(\lambda)$ fit remains relatively good, which indicates that multiple IOP configurations can produce similar reflectance spectra due to the underlying ill-posed nature of the inversion problem. As model complexity increases, the $R_{rs}(\lambda)$ fit improves. The [$k = 3$] reproduces the reflectance spectrum to within 10% at all wavelengths, and the [$k = 4$] and [$k = 5$] models achieve even better agreement, at the several percent level or less. The [$k = 4$] model achieves a reduced chi-squared $\chi_\nu^2 \approx 1$ when assuming 5% uncertainties (S/N = 20) in $R_{rs}(\lambda)$, suggesting that the model complexity is well-matched to the information content in the data.

This progression implies that increasing $k$ improves retrieval fidelity up to a certain point. If $k = 5$ continues this trend, one might expect a reasonably good retrieval of $a_{\mathrm{nw}}(\lambda)$ and $b_{b,p}(\lambda)$ under ideal conditions. However, Figure 5 also underscores a fundamental limitation in IOP retrievals: beyond a certain level of complexity, additional parameters may not significantly





enhance the retrieval unless supported by sufficiently independent spectral information. This illustrates the inherent degeneracy in the inversion problem, where different IOP parameterizations can yield similar $R_{rs}(\lambda)$, making it difficult to extract unique

IOP solutions. Thus, while a $[k=5]$ model might provide a better fit, it remains constrained by the amount of independent spectral information available in the data, reinforcing the challenge of retrieving detailed IOP spectra from multi-spectral or even hyperspectral observations.

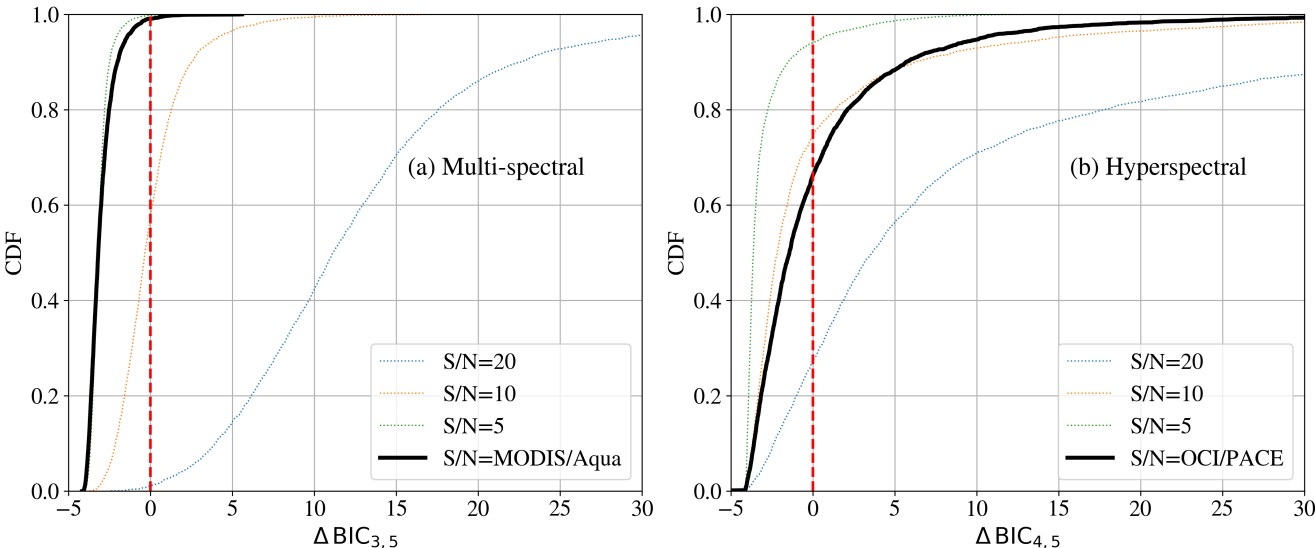

**Figure 6.** Evaluations of the difference in BIC values $\Delta\mathrm{BIC}$ for fits to reflectance data derived from the 3,320 spectra of L23. The curves describe the cumulative distribution function (CDF) of the $\Delta\mathrm{BIC}$ values. The left panel is for simulated MODIS observations (8 bands) with a series of signal-to-noise (S/N) assumptions (colored curves) and for retrievals adopting actual MODIS noise estimates from NASA validation Werdell and Bailey (2002). The $>99\%$ negative $\Delta\mathrm{BIC}_{3,5}$ values for the realistic noise case indicates the L23 spectra greatly favor models with only 3 parameters and without phytoplankton. [right panel] Similar analysis but for OCI/PACE-simulated observations with fixed S/N and for a best guess at nominal OCI/PACE performance (Section 2.3). We find even with OCI that only $\approx 30\%$ of the L23 dataset favors the model with phytoplankton.

We can statistically evaluate the constraining power of the data with the BIC formalism introduced in section 2.1. Figure 6 shows the $\Delta\mathrm{BIC}$ values for simulated MODIS observations of the L23 spectra (see Supp 2.3 for details). The cumulative

distribution function (CDF) on the y-axis represents the cumulative probability that the $\Delta\mathrm{BIC}$ values (Equation 4) for the 3,320 fits to the L23 $R_{rs}(\lambda)$ data are less than or equal to a specific value. Each $\Delta\mathrm{BIC}$ value corresponds to a comparison between two models, specifically the BIC values for a model with and without phytoplankton parameters ($[k=3]$ and $[k=5]$ in the multi-spectral case, Figure 6a, and $[k=4]$ and $[k=5]$ in the hyper-spectral case, Figure 6b). If the CDF value is $y_{\mathrm{CDF}}$ at a specific $\Delta\mathrm{BIC}$ value, it means that the $100 \times y_{\mathrm{CDF}}\%$ of the $\Delta\mathrm{BIC}$ values in the dataset are less than or equal to that specific

value. Thus, the CDF curve shows the proportion of the dataset for which the simpler model is preferred as a function of the



$\Delta$BIC value. The higher the CDF value at $\Delta$BIC $= 0$, the higher fraction of the dataset which favors the simpler model over the more complex one.

For the analysis with a realistic MODIS noise model, fewer than 1% of the spectra prefer the $[k = 5]$ model with phytoplankton. This holds true even though our analysis assumed a perfect forward model and perfect knowledge of the measurement
uncertainties, without correlated errors. Allowing for these uncertainties would result in zero cases with $\Delta$BIC $< 0$. In fact, we find that one cannot retrieve more than 3 parameters from MODIS observations and that even the $[k = 2]$ model is satisfactory for low $Chla$ waters (Appendix B). Without *perfect* knowledge of the absorption by CDOM one cannot retrieve phytoplankton from MODIS observations alone.

The curve corresponding to S/N $= 20$ in Figure 6a shows that while there is some support for the simpler model, indicated
by the CDF values for positive $\Delta$BIC, the more complex model, which includes additional parameters for phytoplankton, is generally preferred. This is because the CDF for negative $\Delta$BIC values is low, indicating that the simpler model is not favored in most of the dataset. In other words, although the simpler model is supported in some cases, the overall trend indicates that the more complex model is usually favored for S/N $= 20$. Thus, reducing noise in the $R_{rs}(\lambda)$ data is essential when increasing model complexity. However, achieving S/N $= 20$ is challenging, even at blue wavelengths in open ocean Case 1 waters, as
demonstrated in numerous validation studies. And, achieving S/N $= 20$ at $\lambda > 600\,\mathrm{nm}$ where absorption by seawater alone is very high may be impossible (Zhang et al., 2022).

We reach even stronger conclusions for simulated SeaWiFS observations which have fewer bands. Unless one identifies an approach to regularly achieve S/N $\gg 10$ measurements in the presence of all error terms (e.g. atmospheric corrections), phytoplankton cannot be retrieved from multi-spectral observations without strong, additional priors. In Appendix B, we examine
the GIOP and GSM models which assume a fixed and steep $S_{dg}$ shape parameter and the negative outcomes of this assumption.

Now consider an assessment with simulated OCI hyperspectral observations on the PACE satellite (see Section 2.3 for details). Our fiducial case uses the L23 spectral sampling and we limit the observations to $400\,\mathrm{nm} < \lambda < 700\,\mathrm{nm}$, outside of which systematics of the L23 dataset and instrumentation dominate the uncertainties in $R_{rs}(\lambda)$, and poor knowledge of the wavelength dependence of the ocean's constituents preclude confident analysis. Figure 6b shows the distribution in the dif-
ference in BIC values, $\Delta$BIC, between the $[k = 4, 5]$ models assuming several choices for the S/N and our estimate for the OCI/PACE noise from v2.0, Level 2 products. We find that OCI/PACE may not recover an assemblage signature of phytoplankton from water with properties similar to those represented by less than half of the L23 dataset (primarily those with lower $Chla$ concentrations). We are led to conclude that one may retrieve four parameters for IOPs from a OCI-like observation and possibly a fifth. Two of these numbers describe the amplitude and shape of $a_{nw}(\lambda)$ parameterized as an exponential and
two numbers describe $b_{b,p}(\lambda)$ modeled as a power-law. Absent strong priors that account for one of these four, extracting even one number describing $a_{ph}(\lambda)$ (at all wavelengths) will be challenging. The following section explores such hyperspectral retrievals in greater depth.





**Figure 7.** IOP retrievals from a BING analysis of a PACE-simulated spectrum (upper left panel) for the $[k = 5]$ model with its standard priors. For this low $Chla$ example (index=175 of the L23 dataset), we recover estimates of $a_{\mathrm{nw}}(\lambda)$ and $b_{b,p}(\lambda)$ and their uncertainties (colored curves and shaded regions which encompass 68% confidence intervals) that are in good agreement with the true values (solid points). However, the 99% uncertainty interval for $a_{\mathrm{ph}}(440\,\mathrm{nm})$ includes vanishingly small values and we would not conclude phytoplankton is detected at even $3\sigma$ significance.

### 3.3 Retrieving $a_{\mathrm{ph}}(\lambda)$ with NASA/PACE

The results in Figure 6b indicate that hyperspectral observations with characteristics representative of data from the NASA/-
PACE mission should have the statistical power to infer the presence of phytoplankton in the majority of ocean waters. With





BING, we may explore further the promise of such $a_{\mathrm{ph}}(\lambda)$ retrievals as well as assess potential biases. This analysis complements the hyperspectral assessment of a portion of the NOMAD dataset of in-situ observations by Erickson et al. (2023).

For the following analysis, we adopt the $[k = 5]$ model and compare results with the GIOP and GSM algorithms re-emphasizing that the latter was only designed for multi-spectral observations and is only included for illustration. Figure 7

shows the results for the $[k = 5]$ model fit to the top spectrum in Figure 5 but now with random noise included and the 68% confidence interval illustrated. The model provides an excellent description of the $R_{rs}(\lambda)$ measurements, but the reduced $\chi_\nu^2$ being much less than 1 suggests potential overfitting of the data. Furthermore, the retrievals are well matched to the known $a_{\mathrm{nw}}(\lambda)$ and $b_{b,p}(\lambda)$ spectra and fully encompassed by the uncertainty. This includes the individual $a_{\mathrm{dg}}(\lambda)$ and $a_{\mathrm{ph}}(\lambda)$ spectra that comprise $a_{\mathrm{nw}}(\lambda)$.

Quantitatively, on the positive side we recover $a_{\mathrm{ph}}(440\,\mathrm{nm}) = 0.0084 \pm 0.0033\,\mathrm{m}^{-1}$ which lies within $1\sigma$ of the correct value ($0.0073\,\mathrm{m}^{-1}$). On the negative side, the uncertainty implies a less than $3\sigma$ detection, i.e. over 1% of the MCMC samples have values $10\times$ lower than the true $a_{\mathrm{ph}}(440\,\mathrm{nm})$. This is due to the degeneracy between $a_{\mathrm{dg}}(\lambda)$ and $a_{\mathrm{ph}}(\lambda)$, as illustrated in Figure 8 which shows a "corner" plot for the 5-parameter model. The very low $a_{\mathrm{ph}}(440\,\mathrm{nm})$ values ($< 10^{-3}\,\mathrm{m}^{-1}$) are correlated with lower (shallower) $S_{dg}$ and higher $A_{dg}$, i.e. a degeneracy between CDOM/detritus and phytoplankton absorption. We also see

from Figure 8 that the $R_{rs}(\lambda)$ offer effectively no constraint on $\beta_{nw}$; its values are almost entirely defined by the prior we have imposed.

Now consider an example with high $Chla$ concentration. Figure 9 presents the BING fit for the $[k = 5]$ model to an example representative of eutrophic (Case II) waters (idx=2773 of the L23 dataset). The resultant $R_{rs}(\lambda)$ spectrum shows the effects of strong detrital and phytoplankton absorption at blue wavelengths, peaking at $\lambda \approx 550\,\mathrm{nm}$. As with the low $Chla$ example, the

$a_{\mathrm{ph}}(\lambda)$ and $a_{\mathrm{dg}}(\lambda)$ retrievals are generally in agreement with their true values aside from an excess of $a_{\mathrm{dg}}(\lambda)$ absorption at the shortest wavelengths. This excess is due to the combination of lower S/N in the $R_{rs}(\lambda)$ data at $\lambda < 450\,\mathrm{nm}$ and errors in the adopted basis functions. In particular, our model does not capture the variations in $b_{b,p}(\lambda)$ at $\lambda < 500\,\mathrm{nm}$ due to phytoplankton and these are compensated (in part) by the higher $a_{\mathrm{dg}}(\lambda)$, yet another manifestation of the $b_b/a$ degeneracy in IOP retrievals. Despite these issues, the resultant $\chi_\nu^2$ is approximately 1 (i.e. the model fits the data well) and a more sophisticated model

introduced to capture the variations in $b_{b,p}(\lambda)$ may result in over-fitting.

We may explore the impacts of imposing priors on model parameters by comparing the fits from GIOP and GSM to this high $Chla$ example. Figure 10 shows the fits to $R_{rs}(\lambda)$ and the IOP retrievals for these 3 models. All three yield a reduced $\chi_\nu^2 \approx 1$ and would be considered acceptable models (GSM is marginal). These results further demonstrate the physical degeneracy within the radiative transfer; here the GIOP model systematically under-estimates both $a_{\mathrm{ph}}(\lambda)$ and $b_{b,p}(\lambda)$ but these compensate

to yield very similar $R_{rs}(\lambda)$ as the $[k = 5]$ model. We also emphasize that the GIOP model is driven to this solution by the strong prior on $S_{dg}$, error in the estimate of $Chla$ from the OC4 algorithm, and the shallower slope for $\beta_{nw}$ estimated from the Lee et al. (2002) prescription. As is obvious from Figure 10, the best-fit $a_{\mathrm{ph}}(440\,\mathrm{nm})$ values vary by a factor of over 400%, much larger than the estimated uncertainties for each. These large variations occur even though the models have statistically acceptable fits. The low $\chi_\nu^2$ values of the $[k = 5]$ and GIOP models further emphasize that the data have limited statistical power

to retrieve any additional on phytoplankton beyond what is described in the Bricaud et al. (1998) prescription. While different





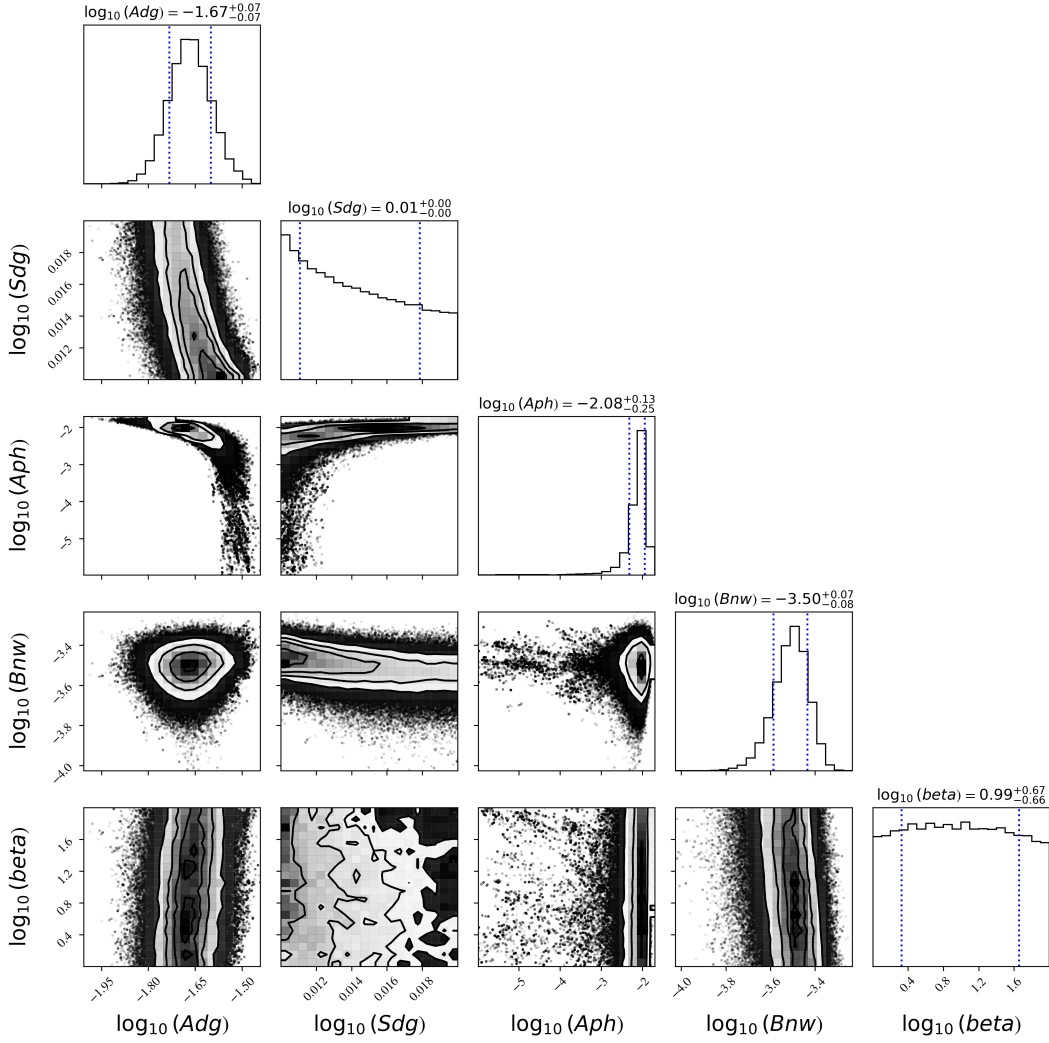

**Figure 8.** A corner plot showing the $\approx 500,000$ samples of the $[k=5]$ model fit shown in Figure 7. The histogram panels show the marginalized posterior distributions of each parameter with the blue dotted lines showing the 68% confidence interval. The contour plots describe the correlations between parameters. Note especially the correlations between $A_{ph}$ and both $A_{dg}$ and $S_{dg}$: models with shallower $S_{dg}$ and higher $A_{dg}$ can describe the data without any phytoplankton absorption. Also note the very poor constraint on $\beta_{nw}$.

retrieval models yield substantially different estimates of $a_{\mathrm{ph}}(440\,\mathrm{nm})$, the spectral information in $R_{rs}(\lambda)$ alone is insufficient to independently resolve phytoplankton absorption without relying on empirical parameterizations. This points to the need for additional observational constraints, such as hyperspectral measurements, to break the degeneracy and improve the accuracy of phytoplankton absorption retrievals.





**Figure 9.** Similar to Figure 7 but for an example spectrum with high $Chla$ concentration and strong detrital absorption. As with the low-$Chla$ case, this fit yields $\chi_\nu^2 < 1$ despite the overestimates of $a_{\mathrm{nw}}(\lambda)$ and $b_{b,p}(\lambda)$ at $\lambda < 450$ nm. This is because of the degeneracy in the radiative transfer ($b_b/a$) and the poorer S/N in $R_{rs}(\lambda)$ at these wavelengths.

We have performed BING fits with the $[k=5]$ and GIOP models to simulated PACE spectra for the full L23 dataset to estimate $a_{\mathrm{ph}}(440\,\mathrm{nm})$ and $b_{b,p}(440)$. As above, we limit to 400-700 nm, adopt the PACE uncertainties (Figure 3, and have injected random noise into each spectrum. Figure 11 compares the $a_{\mathrm{ph}}(440\,\mathrm{nm})$ and $b_{b,p}(440)$ retrievals with BING against the true values. For $a_{\mathrm{ph}}(440\,\mathrm{nm})$, on the positive side, the retrievals track the known values over nearly three orders of magnitude. At low $a_{\mathrm{ph}}(440\,\mathrm{nm})$, however, the $[k=5]$ model systematically under-predicts $a_{\mathrm{ph}}(440\,\mathrm{nm})$ for many of the retrievals leading to a negative bias, and the majority of these are formally upper limits (plotted in a lighter shade of blue). Furthermore, the scatter







**Figure 10.** A series of fits and IOP retrievals for the high-$Chla$ spectrum also shown in Figure 9. The solid, dashed, and dotted curves are the results for the $[k=5]$, GIOP, and GSM models respectively, each with an uncertainty similar to that for the $[k=5]$ model in Figure 9. We find that each model provides a statistically acceptable fit to the $R_{rs}(\lambda)$ data ($\chi^2_\nu \approx 1$) despite the large differences in their IOP retrievals.

is higher than PACE Level 2 requirements (35%; Cetinić et al., 2018). If we limit to cases with true $a_{\mathrm{ph}}(440\,\mathrm{nm}) > 0.01\,\mathrm{m}^{-1}$, the bias is reduced (15%) but the MAE remains large (40%). One of the primary reasons the $[k=5]$ model underestimates low $a_{\mathrm{ph}}(440\,\mathrm{nm})$ values is the spectral degeneracy between $a_{\mathrm{ph}}(\lambda)$ and $a_{\mathrm{dg}}(\lambda)$. When $a_{\mathrm{ph}}(440\,\mathrm{nm})$ is weak, the total absorption is largely dominated by CDOM, making it difficult to separate the phytoplankton contribution from the background absorption (Houskeeper and Hooker, 2025). The $[k=5]$ model allows for flexibility in the spectral shape of $a_{\mathrm{dg}}(\lambda)$, which can lead to overestimation of CDOM/detritus absorption and a corresponding underestimation of phytoplankton absorption.




The GIOP model, in contrast, constrains CDOM absorption using a fixed spectral slope ($S_{dg}$) typically prescribed from empirical studies such as Lee et al. (2002). This constraint prevents the retrieval from assigning excess absorption to $a_{\mathrm{dg}}(\lambda)$ at short wavelengths, reducing the risk of underestimating $a_{\mathrm{ph}}(440\,\mathrm{nm})$. While this approach limits retrieval flexibility, it also helps stabilize the separation between phytoplankton and CDOM absorption, ensuring that even at low $a_{\mathrm{ph}}(\lambda)$ values, the retrieval does not shift excessive absorption to CDOM. The right panels of Figure 11 show the retrievals for $b_{b,p}(440)$. The $[k = 5]$ model provides a better overall agreement with the true values, although some scatter persists, particularly at low $b_{b,p}(440)$ values. The bias and MAE are smaller compared to the GIOP results, suggesting that the additional flexibility in the $[k = 5]$ model helps capture variations in backscattering more accurately. In contrast, the GIOP retrievals exhibit a consistent bias in $b_{b,p}(440)$. This can be attributed to the model's prescribed functional form for backscattering, which lacks flexibility in representing natural variations in $b_{b,p}(440)$ across different water types. The constrained power-law exponent used in GIOP may not accurately reflect regional or case-specific spectral slopes, leading to systematic errors. As a result, while GIOP provides a reasonable fit to $R_{rs}(\lambda)$, its retrieved $b_{b,p}(440)$ values tend to be biased relative to the true values, particularly in optically complex waters.

To improve the Bayesian approach and reduce the underestimation of $a_{\mathrm{ph}}(440\,\mathrm{nm})$ at low values, one may consider refining prior constraints on CDOM absorption, incorporating additional spectral information, and enhancing regularization techniques. Strengthening Bayesian priors on the CDOM spectral slope ($S_{dg}$) based on climatologies or independent datasets can help prevent over-attribution of absorption to CDOM. Incorporating near-UV bands (350–400 nm), where CDOM absorption dominates, provides an additional constraint to improve separation from phytoplankton absorption. Enhancing Bayesian regularization with priors that favor realistic $a_{\mathrm{ph}}(\lambda)$ spectral shapes and implementing adaptive noise weighting can help mitigate retrieval biases in low-absorption regimes. Finally, performing ensemble retrievals, where multiple runs with varied priors are averaged, can further stabilize the retrieval against noise. These refinements will improve retrieval accuracy and reduce systematic underestimation of $a_{\mathrm{ph}}(440\,\mathrm{nm})$ in low-chlorophyll waters.





**Figure 11.** The left panels show the retrievals of $a_{\mathrm{ph}}(440\,\mathrm{nm})$ for simulated PACE spectra against the true values of for the $[k=5]$ (top) and GIOP models (bottom) using the BING package. When the error on $a_{\mathrm{ph}}(440\,\mathrm{nm})$ exceeds three times the best value, we plot the measurement in light blue. The dashed curve is the 1:1 line. The right panels show the retrievals for the particulate backscattering at 440 nm. The bias, meidan absolute error (MAE), and root mean square (RMS) are indicated in each panel.



## 4 Discussion and Future Prospects

In this manuscript, we have introduced BING, a Bayesian inference algorithm for IOP retrievals utilizing the Gordon coefficients for radiative transfer. We have reemphasized a known but under-appreciated physical degeneracy in the radiative transfer – $r_{rs}(\lambda)$ is a function of $b_b/a$ which strictly limits one ability to retrieve $a(\lambda)$ and $b_b(\lambda)$ without strong priors. Two of the priors are natural: water both absorbs and scatters light with precisely known coefficients, at least for wavelengths $\lambda > 400$ nm. We demonstrated, however, that even these constraints are insufficient; indeed, water frequently dominates the model limiting

the extraction of additional information. Consequently, we found that multi-spectral observations with published uncertainties Zhang et al. (2022) cannot reliably retrieve phytoplankton absorption, and that even hyperspectral observations (e.g. OCI/-PACE) will be challenged (Figure 6).

   Previously, Cael et al. (2023) reached a similar inference as one of our primary conclusions – the limited information content of remote-sensing observations. Specifically, they analyzed the degrees of freedom (DoF) of $R_{rs}(\lambda)$ data through a standard

principal component analysis finding that in-situ $R_{rs}(\lambda)$ data with MODIS sampling has only 3 DoF and inferred only DoF=2 for remotely sensed $R_{rs}(\lambda)$. Our analysis, which includes several constraints, such as water absorption and scattering, yields at least one additional parameter but the overarching implication is similar: retrievals from $R_{rs}(\lambda)$ observations have limited information content.

   On statistical grounds, retrieving $a_{\mathrm{ph}}(\lambda)$ from $R_{rs}(\lambda)$ is fundamentally challenging because CDOM and detrital absorption,

which are always present, exhibit strong spectral overlap with phytoplankton absorption in the blue region. In fact, this component ($a_{\mathrm{dg}}$) tends to exceed $a_{\mathrm{ph}}$, even in the open ocean (Siegel et al., 2013; Hooker et al., 2020; Houskeeper and Hooker, 2025). When $a_{\mathrm{dg}}(\lambda)$ is parameterized as an exponential function, small variations in its spectral slope can lead to compensatory shifts in $a_{\mathrm{ph}}(\lambda)$, making it difficult to separate their contributions. Since $R_{rs}(\lambda)$ depends on the combined effects of absorption and backscattering rather than directly measuring individual IOPs, this spectral degeneracy prevents a unique retrieval of $a_{\mathrm{ph}}(\lambda)$

without additional constraints or priors. Previous work that published estimates of $a_{\mathrm{ph}}(\lambda)$ required very strict priors on the shape of $S_{dg}$ (Supp B), leading to significant bias in estimates of $a_{\mathrm{ph}}(\lambda)$.. In the cases where $S_{dg}$ was allowed to vary (Boss & Roesler, Chapter 5 Lee, 2006), the errors on $a_{\mathrm{ph}}(\lambda)$ were severe and limited retrievals only to upper limits, consistent with this work.

   Do our results therefore invalidate the past several decades of research and data products using satellite-based ocean color

observations? At the least, all previous retrievals from $R_{rs}(\lambda)$ must be further scrutinized. We assert uncertainties and biases were frequently (possibly always) underestimated, and substantial correlations between retrieved parameters will be present. Unfortunately, even relative analyses of $a_{\mathrm{ph}}(\lambda)$ may be subject to large error. There are, however, key products that are primarily (and very nearly exclusively) empirical, i.e. generated without any radiative transfer modeling (Stramski et al., 2022). These empirically-based algorithms may circumvent the radiative transfer issues raised here, but as emphasized by Cael et al. (2023),

one cannot retrieve an arbitrary number of such quantities from visible domain $R_{rs}(\lambda)$ observations. Therefore, the suite of products generated by the community to date are highly coupled and correlated. The limited information content of $R_{rs}(\lambda)$ measurements subject to realistic uncertainties is inherent to the problem.





While the results from Figure 6b indicate that hyperspectral observations offer a substantial improvement over multi-spectra data, even *detecting* phytoplankton remains challenging. We reemphasize that the results presented here have assumed a per-fect forward model (i.e. no error in the radiative transfer calculation), uncorrelated uncertainties, a perfect model for water absorption and backscattering, and homogeneous seawater (no vertical or horizontal spatial variations). Furthermore, even if we surmount these issues, we may only be able to extract a single parameter describing phytoplankton, e.g., the $a_{\text{ph}}$ amplitude at $\lambda \approx 440$ nm. And, strictly speaking, this may be attributed to *any* absorption component (not solely $a_{\text{ph}}$) that does not follow the exponential description of CDOM and detritus absorption.

## 5 Conclusions

The ocean color remote sensing community faces a fundamental challenge in retrieving IOPs from remote sensing reflectance due to a physical degeneracy in the radiative transfer equation. Our analysis demonstrates that this degeneracy severely limits the number of parameters that can be reliably extracted from $R_{rs}(\lambda)$ observations. For multi-spectral satellite data with realistic noise levels (e.g., MODIS, SeaWiFS), we find that only three parameters can be reliably constrained, which is insufficient to independently retrieve phytoplankton absorption without strong, potentially biasing priors on the spectral shape of CDOM/de-tritus absorption. Even with hyperspectral observations like those from OCI/PACE, retrievals remain limited to four or five parameters at most, and the detection of phytoplankton absorption is still challenging for many oceanic conditions.

These findings suggest that previous IOP retrieval algorithms likely underestimated uncertainties and may have introduced systematic biases in their estimates of phytoplankton absorption. The widespread practice of fixing the spectral slope of CDOM/detritus absorption ($S_{dg}$) to a steep value in models like GSM and GIOP allows for phytoplankton detection but at the cost of potentially significant biases in $a_{\text{ph}}(\lambda)$ estimates. Our Bayesian approach explicitly incorporates priors and their uncertainties, providing a more transparent and rigorous assessment of the retrieval problem. While hyperspectral observa-tions offer improvement over multi-spectral data, they still cannot fully overcome the fundamental limitations imposed by the physical degeneracy in the radiative transfer equation.

How might we proceed? It is abundantly clear that we must identify the optimal way to parameterize the problem to make most effective use of the 4 or 5 parameters that describe $a_{\text{nw}}(\lambda)$ and $b_{b,p}(\lambda)$. For example, if we know $\beta_{nw}$ (i.e. fix its value as a prior), we would not "waste" a free-parameter to estimate its value. In short, we must harness our knowledge of the ocean from previous in-situ measurements (or current, if one can afford them) to set priors on the model. These priors should be geographically and temporally variable to reflect different oceanic conditions. Because strict and biased priors have been shown to lead to inaccurate and uncertain retrievals, one must proceed cautiously. Empirically derived priors can help mitigate these issues by providing a more accurate representation of the ocean's optical properties.

One obvious community-wide effort would be to develop and agree upon strong priors for $S_{dg}$. This is current practice in many existing algorithms (e.g. GIOP, GSM, which set $S_{dg}$ to a single value), but we describe the negative consequences of this extreme approach in Supp B. Instead, we encourage the community to generate $S_{dg}$ priors as probability distribution functions that vary with geographic location and time and then revisit these in our changing climate. Additionally, we must include more



observations, both from space and in-situ. From space, we must leverage the $Chla$ fluorescence signal at $\approx 685$ nm (Wolanin et al., 2015) whose production and radiative transfer are distinct from that of IOP retrievals. From the ocean, in-situ observations provide invaluable validation data and may establish priors like those for $S_{dg}$. Non-visible in-situ optical observations have been shown to improve retrievals of CDOM absorption, and could be retained to better partition signals related to CDOM and

phytoplankton biomass. Constraints on $\beta_{nw}$, as a function of location and season, and physical priors on the coupling of $a(\lambda)$ to $b_b(\lambda)$ for individual components from the physics of absorption and scattering could be impactful.

Developing community-wide Bayesian retrieval algorithms is also recommended. The ocean color remote-sensing community should be encouraged to adopt a Bayesian framework for IOP retrievals such as BING, explicitly including all priors and their uncertainties. A Bayesian approach allows for a more transparent and rigorous incorporation of prior knowledge and

uncertainties, leading to more reliable retrievals. Unlike BING, however, Bayesian algorithms must adopt an accurate forward model and its uncertainties, include correlated and systematic error in the observations, and harness new data sources. We have initiated such a project – Intensities to Hydrolight Optical Properties (IHOP) – and encourage community adoption and development. The scientists focused on atmospheric corrections have already embarked on this journey following the original insight of Frouin and Pelletier (2015).

The development of machine learning techniques for scientific exploitation is another tempting and potentially viable path to address some of the challenges presented in this manuscript. Optimistically, one can hope to train models that learn correlations in the $R_{rs}(\lambda)$ spectra which predict quantities like $Chla$ or even signatures of phytoplankton communities (e.g. Woo Kim et al., 2022; Kramer et al., 2022). While we appreciate the potential power of such data-driven approaches, one must be mindful of the physical degeneracy of Equation 5 which leads to very similar or even identical $R_{rs}(\lambda)$ spectra from distinct IOPs (e.g.

Figure 10. A machine learning model cannot learn how to distinguish between these, especially in the presence of noise, nor will most industry-developed algorithms properly assess the uncertainties of such degeneracies. In practice, they would behave only according to the datasets they were trained upon; this is an implicit prior (Gray et al., 2024). Lastly, such models will not, on their own, discover new signatures of absorption in the ocean.

Increasing the spectral resolution of satellite observations can provide more detailed information about the absorption and

backscattering properties of phytoplankton, thereby reducing the impact of degeneracies. Thus, the development and deployment of hyperspectral satellites with high spectral resolution across the visible and near-infrared spectrum are recommended. The recently launched PACE satellite will lead the way. Additionally, exploring alternative remote-sensing techniques, such as Lidar and fluorescence-based methods, and incorporating polarization information to complement traditional ocean color observations, should be considered. These advanced techniques may provide independent measurements that help resolve am-

biguities and improve the overall accuracy of phytoplankton estimates, although information gained using these new techniques should be clearly demonstrated and defined first in the field.

Promoting interdisciplinary collaboration is also essential. Fostering collaboration between oceanographers, remote-sensing experts, and radiative transfer modelers to address the complex challenges of IOP retrievals can bring together diverse expertise and perspectives, leading to more innovative and effective solutions. These should include individuals with mastery of statistics

who can rigorously assess uncertainty and help develop robust and transparent algorithms.



By implementing these recommendations, the remote-sensing community can significantly enhance the accuracy and reliability of phytoplankton IOP retrievals, leading to better-informed biogeochemical models and ecological assessments. This comprehensive approach will help ensure that remote-sensing data accurately reflect the true state of the ocean's biological and chemical processes, thereby supporting more effective environmental monitoring and management efforts.

*Code and data availability.* All of the code and results presented here are available on GitHub in these two repositories: https://github.com/ocean-colour/bing, https://github.com/ocean-colour/ocpy. The data on which this article is based are available in Loisel et al. (2023).

**Appendix A: Scrutinizing the Taylor series expansion approximation to the radiative transfer**

To examine the use of Equation 5 as an approximation of the radiative transfer, consider Figure A1 which plots the Hydrolight-derived $R_{rs}(\lambda)$ of L23 converted to $r_{rs}(\lambda)$ with Equation 5 against the evaluated $u(\lambda)$ values using Equation 6 at four distinct

wavelengths. These evaluations approximately follow a quadratic with zero y-intercept. Overplotted with a dashed line is the Gordon approximation using the standard $G_1, G_2$ coefficients and Equation 5. Qualitatively, the Hydrolight outputs follow the relation yet lie systematically above the curve. At its extreme, the Taylor series approximation is offset by $\approx 10\%$ at $\lambda = 370$ nm and $u(\lambda) = 0.35$.

To further illustrate the difference, we have fitted $G_1, G_2$ coefficients to the data at select wavelengths and recover similar $G_1$

values but $G_2$ values that vary significantly with wavelength ($G_2 \approx 0.07$ at $\lambda = 370$ nm, $G_2 \approx -1.2$ at $\lambda = 600$ nm). We also find that there is significant scatter around each of the fits with a relative RMS of $\approx 5\%$ at shorter wavelengths and $20\%$ at the reddest wavelengths. We expect this scatter is inherent to Equation 5 and would be unavoidable if one uses this approximation even with wavelength-dependent coefficients. An accurate retrieval algorithm would need to account for these variations or otherwise suffer this systematic error. This is the focus of a separate algorithm we are developing, and we also refer the readers

to recent advances in approximations of the radiative transfer equation (Twardowski et al., 2018).



## Appendix B: Revisiting Previous Models

The simple IOP models examined in Section 3.2 resemble prescriptions adopted previously in the literature and/or implemented operationally by NASA. In light of our results, we are motivated to further examine two such models: GSM and GIOP. As described in Section 2.5, both GSM and GIOP adopt strict (fixed) priors on $S_{dg}$ and $\beta_{nw}$ such that these are effectively 3-
parameter models.

We find that by adopting these priors, these 3-parameter GSM and GIOP models are statistically favored ($\Delta \mathrm{BIC} > 0$) relative to the $[k=3]$ model without phytoplankton for $\approx 50\%$ (GSM on SeaWiFS) or more (GIOP on MODIS) of the simulated spectra (Figures A2 and A3). At the same time, if we relax either the prior on $S_{dg}$ or $B_{nw}$, over 99% of the spectra have $\Delta \mathrm{BIC} < 0$ and favor the $[k=3]$ model.

Furthermore, the fixed $S_{dg}$ value adopted in each of these models is relatively steep, which imposes a significant bias on the $a_{\mathrm{ph}}(\lambda)$ retrievals. Let us scrutinize these priors as they affect the potential to retrieve phytoplankton and any other constituents. Figure A4 shows the $S_{dg}$ values derived with the $[k=4]$ model (no phytoplankton) against the fraction of non-water absorption associated with phytoplankton at 440 nm in the L23 spectra, $a_{\mathrm{ph}}/a_{\mathrm{nw}}$. The two quantities are anti-correlated because the increased presence of phytoplankton relative to CDOM and detritus tends to give a shallower, non-water absorption
spectrum. We find, as anticipated, that the majority of retrieved $S_{dg}$ values lie within the loci of shape parameters assumed by L23 for CDOM and detritus based on (Lee, 2006). There is, however, a non-negligible set of retrieved $S_{dg}$ values that are lower than the lowest value assumed by L23; these are partially due to strong phytoplankton absorption. Overplotted on the figure are the fixed values of $S_{dg}$ for the GSM and GIOP models where we see their priors lie at the upper end of $S_{dg}$ values measured in the ocean (GIOP) or even beyond (GSM). This was intentional for GSM (Maritorena and Siegel, 2005), as its
designers derived fixed values for $S_{dg}$ and $\beta_{nw}$ by achieving best retrievals when compared against in-situ data. When one adopts a relatively steep $S_{dg}$ value, the absorption at $\lambda > 450$ nm cannot be correctly described by CDOM/detritus and the model will favor a higher phytoplankton contribution. If, however, the $S_{dg}$ values are too steep, then one may anticipate biased $a_{\mathrm{ph}}(440\,\mathrm{nm})$ retrievals.

We then performed a new set of inferences on the entire L23 dataset assuming MODIS and SeaWiFS simulated spectra
for the GIOP and GSM models respectively. In both cases, we calculated $R_{rs}(\lambda)$ from Equations 5 and 7 and performed the inversion with the same model after perturbing the $R_{rs}(\lambda)$ values due to the presence of noise. The retrievals are presented in Figures B1 and A5. Clearly, the retrievals of $a_{\mathrm{ph}}$ and $b_{b,p}$ are biased and highly uncertain at all values, with nearly two orders of magnitude of scatter. Therefore, the detection of $a_{\mathrm{ph}}(\lambda)$, if it were possible with multi-spectral observations, would be highly uncertain.

The constraints inherent in inversion algorithms like GSM and GIOP do affect the confidence in interpreting changes in maps of retrieved variables such as chlorophyll concentration, absorption coefficients, and backscattering coefficients. The spectral ambiguity in $R_{rs}(\lambda)$ data can lead to changes influenced by variations in other optical properties not fully addressed by the models, making it difficult to attribute changes solely to biological factors. Moreover, the interdependence of retrieved parameters, such as chlorophyll, $a_{\mathrm{CDOM}}$, and $b_{b,p}$, means that errors in one can propagate to others, complicating the interpretation



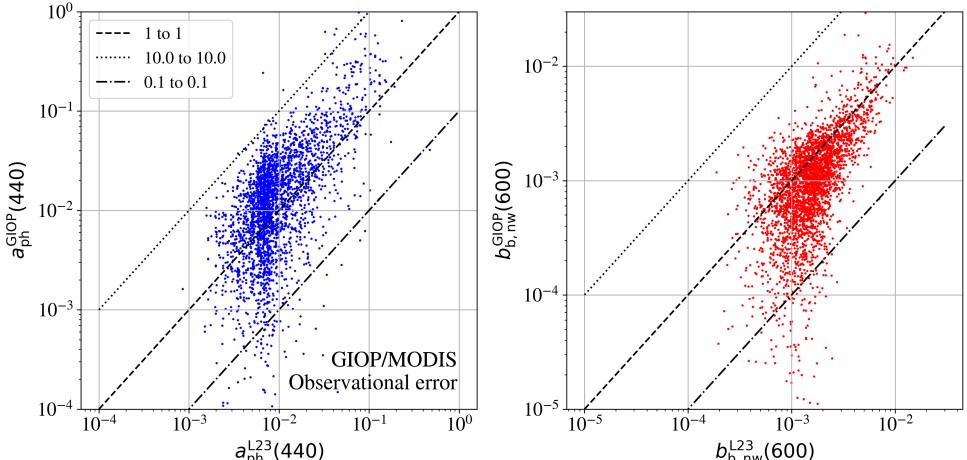

**Figure B1.** Retrievals of $a_{\mathrm{ph}}(440\,\mathrm{nm})$ and $b_{b,p}$ at $600\,\mathrm{nm}$ for the GIOP model with simulated MODIS spectra and perturbing the $R_{rs}(\lambda)$ values by the typical noise. We find the values are biased and scatter by an order of magnitude or more.

of these maps. For example, inaccuracies in backscatter coefficient estimates can affect chlorophyll retrievals. Additionally, variability in environmental conditions can impact the accuracy of the retrieved variables. Algorithm performance may vary across different water types and regions, necessitating further caution in interpreting these changes.

Results from tests of IOP retrieval models have been presented in multiple publications over the years since the beginning of their development (e.g. Mouw et al., 2017; Werdell et al., 2018; Seegers et al., 2018), and a discussion of all of these lies beyond

the scope of this manuscript. Nevertheless, we wish to highlight one, in-depth effort summarized by the IOCCG Report 5 (Lee, 2006). Similar to our work, the participants applied their IOP retrieval algorithms to a simulated (i.e. known) dataset to assess performance. The majority of these algorithms assumed an exponential term for CDOM/detritus absorption with fixed $S_{dg}$ and a steep value ($S_{dg} > 0.015\,\mathrm{nm}^{-1}$). Similar to the results we found for GIOP and GSM (Figure A5,B1), these consistently over-estimated $a_{\mathrm{ph}}(\lambda)$ at $440\,\mathrm{nm}$.

Only one team (Boss & Roesler) allowed $S_{dg}$ to vary (from 0.008 to 0.023 $\mathrm{nm}^{-1}$) in their algorithm which followed from the Roesler et al. (1989) publication. Referring to their Figure 8.1, one notes less biased $a_{\mathrm{ph}}(\lambda)$ values than the other algorithms and that they were the only group to include an error estimation. Given the axes are a log-scaling, one might miss that the uncertainties in $a_{\mathrm{ph}}(\lambda)$ are large enough to be consistent with zero. This implies that the retrieved values are not statistically distinguishable from zero at the given confidence level. In other words, the results from the only algorithm that allowed $S_{dg}$ to

vary freely indicate that $a_{\mathrm{ph}}(\lambda)$ could not be reliably constrained from the simulated data.





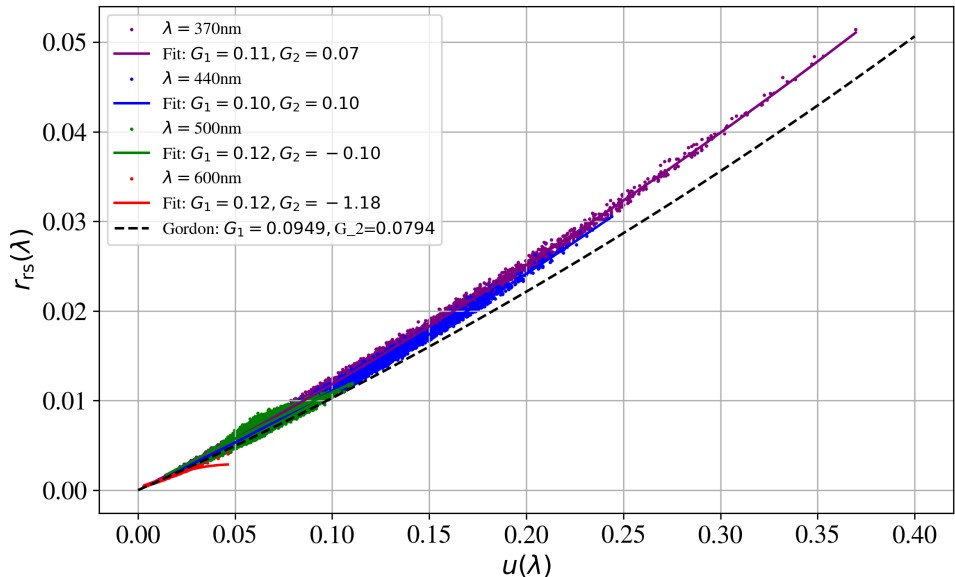

**Figure A1.** Sub-surface reflectances $r_{rs}(\lambda)$ generated with the Hydrolight radiative transfer code by L23 (converted from $R_{rs}(\lambda)$ using Equation 7) against $u(\lambda)$ as defined by Equation 6. The black dashed line shows the second order Taylor expansion of $r_{rs}(\lambda)$ in $u(\lambda)$ (Equation 5) with the Gordon coefficients most widely adopted by the community. In general, this curve underpredicts $r_{rs}(\lambda)$ as calculated with Hydrolight with a maximum offset of $\approx 10\%$ at $u(\lambda) = 0.35$ and $\lambda = 370$ nm. We also show a series of individual fits of Equation 5 to the data with the legend indicating the derived $G_1, G_2$ coefficients. Note that $G_1$ is largely independent of wavelength but that $G_2$ is strongly wavelength dependent (and anti-correlated).





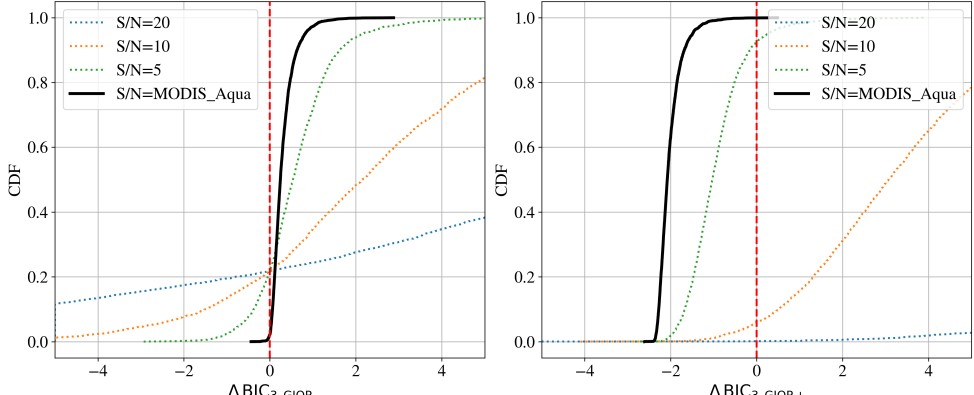

**Figure A2.** These panels describe the BIC analysis assuming MODIS-like observations for two models designed to match standard GIOP configurations. The (left) panel compares our $[k=3]$ model against the standard $k=3$ parameter configuration of GIOP (see text for full details). We find this GIOP model is preferred, which we speculate is due to the extra freedom to fit the $R_{rs}(\lambda)$ at blue wavelengths where the S/N in MODIS is highest. (right) Results for the GIOP+ model ($k=4$ parameters) which lets $\beta_{nw}$ be an additional free parameter. This model is disfavored for the entire dataset, further evidence that one cannot recover 4 parameters from MODIS observations.



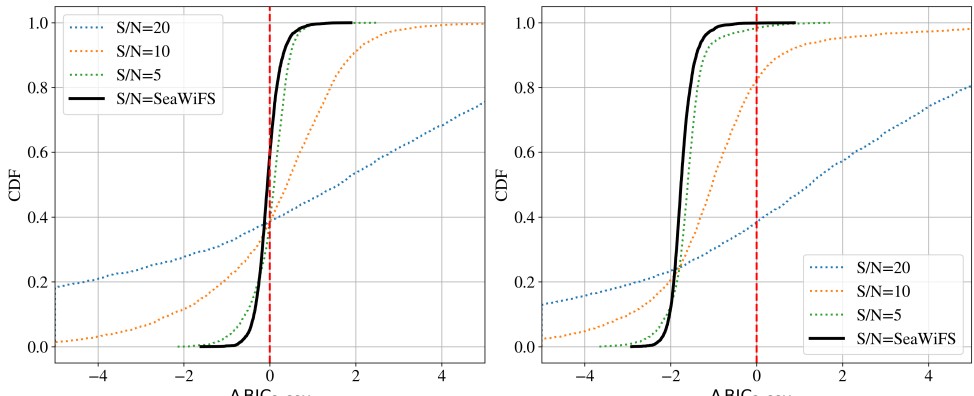

**Figure A3.** Similar to Figure A2 but for GSM models and using simulated SeaWiFS spectra. As with the GIOP models, we find the GSM model is favored over our $[k = 3]$ model but that a $k=4$ parameter version – GSM+ which lets $\beta_{nw}$ be free – is highly disfavored.





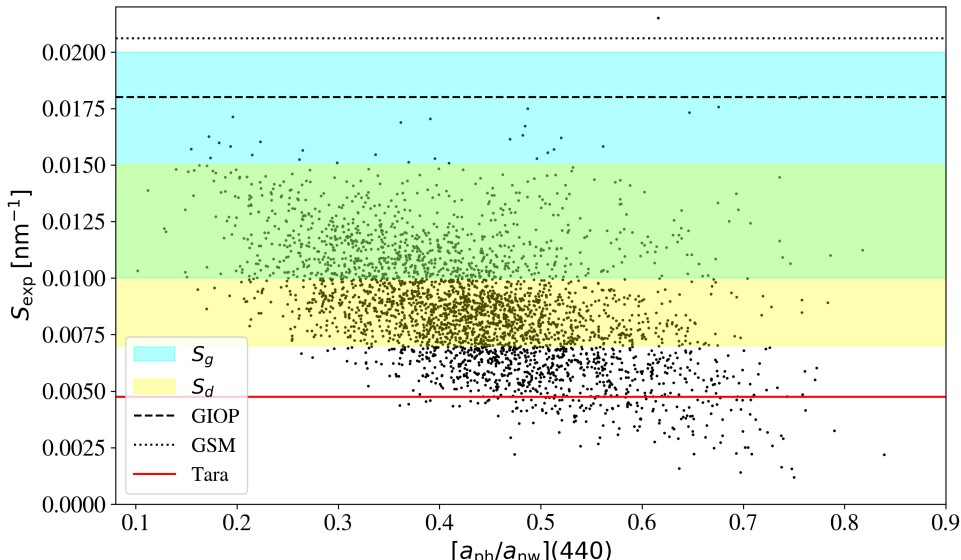

**Figure A4.** The dots plot the best fit shape parameter $S_{dg}$ for the L23 spectra using the $[k = 4]$ model (no phytoplankton component) versus the amplitude of $a_{\mathrm{ph}}$ to $a_{\mathrm{nw}}$ at 440 nm. The two are correlated, albeit with large scatter. The blue and yellow shaded regions indicate the ranges of exponential shapes for CDOM and detritus ($S_g, S_d$) respectively adopted by L23. Not surprisingly, the majority of retrieved $S_{dg}$ lie within these loci. The ones with shallower slope, however, may be attributed to the presence of phytoplankton which effectively flattens $a_{\mathrm{nw}}(\lambda)$ at $\lambda \approx 450$ nm. The black dotted/dashed lines demarcate the fixed values of $S_{dg}$ assumed by the GIOP/GSM algorithms for $a_{\mathrm{dg}}(\lambda)$. These are steeper values than the typical $S_g$ (and all $S_d$) values adopted by L23. In addition, we shows the $S_{dg}$ derived from an extreme, CDOM-subtracted *Tara* absorption spectrum collected off the coast of Africa (see Prochaska and Gray, 2024) which demonstrates at least one instance of a very low $S_{dg}$ in the ocean.



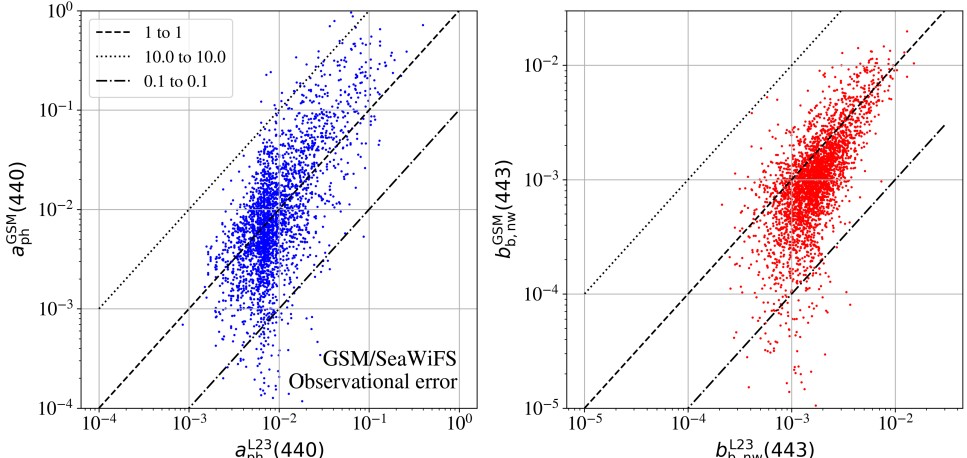

**Figure A5.** Same as Figure B1 but for the GSM model and simulated SeaWiFS spectra.



*Author contributions.* JXP conceived of the paper, generated all of the code for BING and all of the figures of the manuscript. He led the writing. RF provided scientific guidance, text, and extensive editing.

*Competing interests.* The authors declare no conflicts of interest.

**Financial Support**

JXP acknowledges support from a Simons Pivot Fellowship. RF was supported by the National Aeronautics and Space Administration under various grants.

*Acknowledgements.* The authors wish to thank B. Ménard, E. Boss, S. Kramer, A. Windle, H. Housekeeper, and C. Mobley for discussions and/or their input on an earlier draft. Last, we acknowledge deriving new insight on the problem from conversations with M. Kehrli.





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
