# Peer review of "On the Challenges of Retrieving Phytoplankton Properties from Remote-Sensing Observations"

_EGUsphere, 2025_

## Author Response (AR1)

**Final Response**

We thank RC1 for their careful reading of the manuscript and their comments and criticism. Below, we detail the changes we plan to make in the revised manuscript, provided Editor Suzuki allows the review to proceed. Our responses are indicated by the >> prefix.

Minor Comments and Editorial Suggestions

Given the centrality of Rrs to the analysis, the definition at Lines 26–28 would benefit from being presented as a standalone, numbered equation to highlight the role of the bb/a ratio. This would aid readers less familiar with radiative transfer.
>> This is an excellent suggestion, and we have added individual equations defining each of Rrs, a and bb.

Paragraphs beginning at Lines 70 and 75 end with a repeated sentence, this should be revised for clarity.
>> We corrected this editing mistake.

Line 211: possibly intended to read "For the principal analysis"?
>> We would correct this spelling error.

Fig. 4 caption: revise "spectrum that maintaine" to "spectrum that maintains."
>> We corrected this typo.

Several paragraphs in the Results section, such as Lines 269–274 and parts of the Fig. 4 caption, describe experimental setup rather than findings. To follow standard structure, this content should be moved to the Methods section. For example, the description of parameter scaling or idealized simulation conditions belongs in Methods rather than the figure caption.
>> This is an excellent suggestion and we added it to our Methods section. In particular, we introduced a new "arbitrary IOP model" and defined it as described in the Fig 4 caption.

In Fig. 6, consider increasing line thickness for the S/N 5–20 curves to improve visibility.
>> We prefer to keep this figure as is to keep the emphasis on the actual, estimated S/N of the sensors.

Lines 339 and 349 reference "Figure 6b," but figure panels are not labeled. Either label panels (a, b, c) or refer to them using a nomenclature (e.g., left, right) consistent with the captions.
>> We edited the Figure caption.

Spelling: "Retrieval" is misspelled in some of the Fig. 6 legends.
>> We corrected this mis-spelling.

> For Fig. 8, using a non-monochromatic color scale would improve contrast and
> interpretability of parameter correlations.
> **>>** We experimented with other colors, but have found black to be best.

We thank RC2 for their careful reading of the manuscript and their comments and criticism. Below, we detail the changes we plan to make in the revised manuscript, provided Editor Suzuki allows the review to proceed. Our responses are indicated by the >> prefix.

The authors offer several practical strategies for improving retrieval accuracy, such as incorporating near-UV bands to better constrain CDOM, refining priors on the spectral slope and adopting adaptive regularization techniques. These are sensible and actionable, though the paper would benefit from a more specific discussion of how these approaches might be implemented in operational contexts.

>> This is a very helpful comment. We have added several paragraphs at the end of the discussion on this topic.

While the novelty of the approach is incremental considering prior Bayesian work, the paper's systematic model hierarchy, open-source implementation, and discussion of retrieval challenges make it a meaningful contribution. Its conclusions are both technically sound and of high relevance to the ocean color community, particularly in the context of PACE and future mission planning.

With minor editorial revisions and some structure reorganization around model implementation, this manuscript will serve as a valuable reference for researchers and developers seeking to advance ocean color retrieval methodologies.

**>>** Thank you!

Major comments:

The scientific approach, methods and results are well explained/presented, and their quality seem sufficiently high for the relevant scientific communities. The Authors' conclusion (described above) is also clear to Readers, and the manuscript seems timely to present, too, when one considers ocean color missions on-going or planned worldwide. Overall, the manuscript is generally well-written.

Although the scientific question raised in this manuscript was previously investigated and the conclusion derived from the present manuscript was also similar to the previous work

as the Authors also state it in the manuscript, the present manuscript delivers, using a method different from the previous work, detailed insights of the scientific problems more than just providing the result that only a few parameters can be extracted from Rrs independently. Especially, the Authors demonstrate, using Bayesian approach, how complexity of the bio-optical modelling / parameterization impact on the ocean color retrievals. This helps the Readers to better understand the scientific problem behind the ocean color remote sensing, adding further pedagogical values to the manuscript. As the result, the results presented by the manuscript is worth to be shared among the relevant scientific communities and it would contribute to developing a new ocean color algorithms in a non-conventional manner.

**>>** Thank you!

Minor comments

L83: The Authors mention that "a Bayesian framework leverages well developed technique to assess error and correlations in the results without requiring Gaussianity, i.e. the assumption that errors, uncertainties or distributions of retrieved parameters follow a Gaussian distribution". I understand that this is a very general description. However, the Authors actually assume Gaussian distribution in Rrs and its uncertainties in this manuscript (L123-L125), so the above statement is not appealing.

**>>** This is a fair criticism, although we note that BING does allow for non-Gaussian errors. And once PACE provides a full correlation matrix of their uncertainties, we will incorporate them. We added text to this effect to the revised manuscript, as a footnote.

L179-186: Y-axis label in Figure 2 is misleading if it represents the "simulated Rrs".

**>>** Both axes are measured Rrs. Neither is simulated.

L197-209: Please plot a relative error (or signal to noise) in Figure 3 as an additional information for Readers to better understand the Authors' analysis and discussion.

**>>** We added an additional curve showing a nominal S/N estimate for a fiducial Rrs spectrum.

L215-L245: I would suggest the Authors to replace the CDOM component in Eqs. 11 and 13 by phytoplankton component using Eqs. 17 and 18, otherwise add such a case for k=2, because it corresponds to the so-called Case I water historically and extensively investigated by the ocean color community.

**>>** This is a clever suggestion.  We have implemented this idea, and find that it does not capture well the absorption by CDOM and therefore generally yields poor models.  We added text to this effect in the revised manuscript.

Eq. 15: Aph*aph(lambda) should read aph(lambda).

L257: Equation 15 should read Equation 17.

**>>** We fear we confused matters by using Aph twice.  We clarified this in the revised version.  In particular, we have used Cph instead of Aph in Equation 20 (previously 17).

Figure 4: Please add the figure legend for each dotted curve with a matching scaling factor of the non-water absorption (0.9, 3., 10, 100).

**>>** We added a Legend.

L321-335: These are very important results reflecting the Authors' conclusion, as written in Abstract, that "multi-spectral satellite observation lack the statistical power to recover more than tree parameters describing no-water absorption and backscattering". Therefore the full details could have been described in the main text, not in the Appendix B.

>> We attempted to move at least one of the figures from the Appendix but found that it was too disruptive to the text.  We have instead added an additional paragraph and further links to the Appendix.

L341: I would wonder why the assemblage signature of phytoplankton is suddenly described here? This would confuse the Readers. Perhaps, the Authors discuss about it in Discussion, if desired.

**>>** This was a typo and should have been "absorption".

L343-344: The Authors conclude that one may retrieve four or five parameters. Since the k=5 is the upper limit of complexity set in the Authors' experiments, the Reader would wonder what happens if the Authors further increase the complexity to, say, k=6 (e.g. Eqs16 and 17 without using Eq. 18). Can six (or five) parameters be retrievable (when S/N is set adequately) ?

**>>** We have considered a 6 parameter model where the shape for phytoplankton follows Bricaud but is separate from the normalization (similar to one of the variants of GIOP). While this provides slightly better first, the extra degree of freedom is not favored, i.e. the k=4 model yields lower BIC values in nearly every case.  This is even true for S/N = 20.  We decided not to add text on this in the manuscript.

L468: In nature, some variables are related unavoidably. For example, phytoplankton both absorb and scatter light, so a parameter in phytoplankton absorption may be correlated with a parameter in phytoplankton backscattering, or even a total scattering when phytoplankton dominate. In fact, there would be a natural correlation even among different variables as shown in Figure A4. If I did not misunderstand, the Authors describe the number of "statistically-independent" parameters derivable from the ocean color measurements. Is this correct understanding? Regardless of the answer, I would suggest the Authors to clarify or emphasize that point, to avoid the possible misinterpretation of the Authors' conclusion by the Readers.

>> This is an excellent point, and an aspect the authors have been considering as well.  We are unaware of an "easy" way to manifest any such prior but will explore it further in future work.  But, in any case, we have added text to the revised manuscript on this matter at the end of the Discussion section.

L468-471: Did the Authors mean that "the number of parameters" is same as, or equivalent to "the information content"?  Can the number of parameters be also the number of variables if a spectral model of a variable is parameterized by a single parameter?

**>>** They are effectively equivalent and we meant them as such.  We have clarified the text accordingly.

L526: I wonder if the Authors' result and conclusion may change when the inelastic scattering is considered. The Authors may want to make a comment about it here.

**>>** This is a good question.  At the most basic level, applying the Gordon approximation to real Rrs (as is often the case) will imply a poor model and this should generally lead to poorer results.  It is possible, however, that we may be able to model the inelastic scattering and thereby gain statistical power on the IOP model.  We added text to this effect in the revised version.

—-

Here are a few additional edits we made:

1. We removed the sentence on the X=4,Y=0 model (near Line 160) as it wasn't relevant and may confuse the reader.

---

## Author Response (AR2)

**Final Upload**

We have modified the final manuscript as follows:

1. Removed all of the bold font that previously indicated the changes in our revised manuscript.
2. We have included the full first name of each author on the Title page.
3. Modified the color scheme of figure A1 to be sensitive to color blindness.

We thank you for the support in the publication process.